# Domain Transfer Becomes Identifiable via a Single Alignment

**Sagar Shrestha** [1]  **Subash Timilsina** [1]  **Hoang-Son Nguyen** [1]  **Xiao Fu** [1]

## Abstract

Domain transfer (DT) maps source to target distributions and supports tasks such as unsupervised image-to-image translation, single-cell analysis, and cross-platform medical imaging. However, DT is fundamentally ill-posed: push-forward mappings are generally non-identifiable, as measure-preserving automorphisms (MPAs) preserve marginals while altering cross-domain correspondences, leading to content-misaligned translation. Recent work shows that MPAs can be eliminated by jointly transferring multiple corresponding source/target conditional distributions, but supervision signals labeling such conditionals are not always available in practice. We develop an alternative route to DT identifiability. Under a structural sparsity condition on the Jacobian support pattern, we show that distribution matching together with a single paired anchor sample suffices to identify the ground-truth transfer—requiring substantially less supervision than prior approaches. To enable practical high-dimensional learning, we further propose an efficient Jacobian sparsity regularizer based on randomized masked finite differences, yielding a scalable surrogate without explicit Jacobian evaluation. Empirical results on synthetic and real-world DT tasks validate the theory.

## 1. Introduction

Domain transfer (DT) aims to convert a source distribution to a target distribution, using samples from both distributions without aligned sample pairs (Zhu et al., 2017; Liu et al., 2023; Courty et al., 2016). A widely adopted idea for DT is to learn a neural network-represented transfer functions by matching probability distributions of the target domain and the transformed source domain (Zhu et al.,

2017; Choi et al., 2020; De Bortoli et al., 2024). Many popular learning paradigms in machine learning, including CycleGAN (Zhu et al., 2017) and flow matching (Liu et al., 2023), use this insight.

### 1.1. Domain Transfer and Non-Identifiability

DT enjoys empirical success in many applications, e.g., transfer learning (Zhuang et al., 2020), domain adaptation (Ganin et al., 2016; Courty et al., 2016; Zhang et al., 2019), image-to-image translation (Zhu et al., 2017; Choi et al., 2020; De Bortoli et al., 2024), language and speech translation (Conneau et al., 2018; Lample et al., 2018; Baevski et al., 2021), etc. Nonetheless, it is well-known that DT can learn multiple solutions and may produce undesired effects in applications (Galanti et al., 2018; Moriakov et al., 2020; Shrestha & Fu, 2024). The reason is subtle but critical: Assume that the target domain is generated by a "content-preserving" transfer function from the source domain. Such a function maps the source samples (e.g., handwritten digits) to the target domain (e.g., printed digits) without losing its semantic meaning (the identity of the digits like "2" and "9")—which is often the most useful one to learn in applications. However, distribution matching does not uniquely recover the content-preserving transfer function, due to the existence of *measure-preserving automorphism* (MPA) (Galanti et al., 2018; Moriakov et al., 2020; Shrestha & Fu, 2024). When the data distribution admits an MPA, which is generally the case for continuous distributions (Shrestha & Fu, 2024), multiple spurious solutions exists that can perfectly match the distribution but completely alter cross-domain correspondences—leading content-misaligned transfers (e.g., translating handwritten 2 to printed 9).

The non-identifiability of the content-preserving map has been long known in ML and had been mostly tackled by empirical solutions (Park et al., 2020; Xu et al., 2022; Taigman et al., 2017). A recent line of work provided a theory-backed solution, namely, *diversified distribution matching* (DDM) (Shrestha & Fu, 2024; 2025), to establish transfer map identifiability: instead of matching only the full marginals of source and target distributions, one matches multiple pairs of cross-domain conditional distributions induced by auxiliary variables/attributes. DDM effectively eliminates MPAs and yields identifiability of the content-preserving map under

[1]School of Electrical Engineering and Computer Science, Oregon State University, Corvallis, Oregon, USA. Correspondence to: Xiao Fu <xiao.fu@oregonstate.edu>.

*Proceedings of the 43rd International Conference on Machine Learning*, Seoul, South Korea. PMLR 306, 2026. Copyright 2026 by the author(s).

reasonable conditions (Shrestha & Fu, 2024; 2025). However, this approach requires nontrivial side information (e.g., labels or attributes serving as conditional distribution allocation indicators) for all samples, which may be unavailable or costly to obtain in some DT applications.

## 1.2. Contributions

To address this challenge, this work puts forth an auxiliary information-reduced route to DT identifiability—requiring only as few as *a single* paired "anchor" sample. Our detailed contributions are as follows:

*(i) DT Identifiability via a Single Alignment.* We study a DT setting in which the target domain is generated from the source via a content-preserving transformation. By revealing a connection between DT and nonlinear unmixing, we establish identifiability of the ground-truth transfer function. Specifically, we reinterpret DT as a nonlinear unmixing problem in which source-domain samples act as visible latent components. We show that the presence of a single aligned source–target sample pair suffices to resolve the intrinsic ambiguities of nonlinear unmixing, enabling exact identification of the content-preserving transfer map. Building on recent advances in sparse nonlinear unmixing, we further show that such identifiability can be realized via imposing a mild structural sparsity condition on the Jacobian support of the transfer function. Compared to prior work that relies on richer auxiliary supervision (e.g., conditional distribution labels for each sample) (Shrestha & Fu, 2024; 2025), our result substantially reduces the amount of alignment information required—demonstrating that even a single aligned pair can be sufficient in practice.

*(ii) Efficient Implementation for High-Dimensional Problems.* Existing Jacobian-sparsity–based nonlinear unmixing methods have primarily been validated in low-dimensional settings and rely on analytical sparsity-promoting regularizers (e.g., $\ell_1$ norm and variants) (Zheng et al., 2022). However, directly enforcing Jacobian sparsity becomes prohibitive in high dimensions, as explicit Jacobian computation requires as many back-propagation passes as the input dimension. To address this challenge, we introduce an efficient Jacobian sparsity regularizer based on random vector-masked finite differences, which provides a scalable surrogate without explicit Jacobian extraction. We show that this regularizer closely approximates the original Jacobian sparsity objective under reasonable conditions. When combined with adversarial distribution matching and a simple anchor loss, the resulting formulation yields a practical, end-to-end learning objective suitable for high-dimensional domain transfer.

We validate the proposed method empirically on synthetic and real-world data, demonstrating improved content alignment over standard distribution matching baselines.

**Notation.** Let $x$, $\boldsymbol{x}$, $\boldsymbol{X}$, and $\mathcal{X}$ denote a scalar, vector, matrix, and a set, respectively. Let $p_{\boldsymbol{x}}$ denote the *probability density function* (pdf) of random vector $\boldsymbol{x}$. For a given function $\boldsymbol{g}$, let $\boldsymbol{g}_{\#p_{\boldsymbol{x}}}$ denote the push-forward of $p_{\boldsymbol{x}}$ via $\boldsymbol{g}$, informally the pdf of $\boldsymbol{g}(\boldsymbol{x})$, where $\boldsymbol{x} \sim p_{\boldsymbol{x}}$. Let Id denote the identity function with appropriate domain. $\|\cdot\|_0$ and $\|\cdot\|_1$ denote entry-wise $\ell_0$ and $\ell_1$ norms of given vector or matrix, respectively. Let $\mathrm{supp}(\mathcal{X}) = \{(i,j) \mid \exists \boldsymbol{X} \in \mathcal{X}, X_{ij} \neq 0\}$ and $\mathrm{supp}(\boldsymbol{X}) = \{(i,j) \mid X_{ij} \neq 0\}$ denote the indices of the support of set $\mathcal{X}$ (a set of matrices) and matrix $\boldsymbol{X}$, respectively. For a set of index pairs $\mathcal{I} \subseteq [D] \times [D]$, let $\mathcal{I}_{i,:} = \{j \mid (i,j) \in \mathcal{I}\}$ and $\mathcal{I}_{:,j} = \{i \mid (i,j) \in \mathcal{I}\}$. Let $\boldsymbol{f}(\mathcal{X}) = \{\boldsymbol{f}(\boldsymbol{x}) \mid \boldsymbol{x} \in \mathcal{X}\}$.

## 2. Background

### 2.1. Domain Transfer and Identifiability

Let $\boldsymbol{x} \sim p_{\boldsymbol{x}}$ and $\boldsymbol{y} \sim p_{\boldsymbol{y}}$ denote the random variables representing the data distribution of two simply connected open domains $\mathcal{X}$ and $\mathcal{Y}$, respectively. We consider the case where the $\boldsymbol{y}$-domain is generated from the $\boldsymbol{x}$-domain via a deterministic, smooth, and invertible ground-truth map $\boldsymbol{g}^{\star} : \mathcal{X} \to \mathcal{Y}$ as a result of the following data generating process:

$$\boldsymbol{x} \sim p_{\boldsymbol{x}}, \qquad \boldsymbol{y} = \boldsymbol{g}^{\star}(\boldsymbol{x}), \tag{1}$$

In push-forward notation, this is equivalent to $p_{\boldsymbol{y}} = [\boldsymbol{g}^{\star}]_{\#} p_{\boldsymbol{x}}$. In applications, this $\boldsymbol{g}^{\star}$ represents the desired content-preserving map.

Note that we are interested in the *unsupervised* domain transfer setting—that is, only samples of $p_{\boldsymbol{x}}$ and $p_{\boldsymbol{y}}$ are accessible, but the aligned pairs $(\boldsymbol{x}, \boldsymbol{g}^{\star}(\boldsymbol{x}))$ are not available.

Our goal is to provably identify $\boldsymbol{g}^{\star}$ under this setting via designing a learning criterion. Formally, we use the following definition:

**Definition 2.1** (Transfer Map Identifiability)**.** A learning criterion (or a learning loss) identifies $\boldsymbol{g}^{\star}$ from $\boldsymbol{x} \sim p_{\boldsymbol{x}}$ and $\boldsymbol{y} \sim p_{\boldsymbol{y}}$ under (1) if the optimal $\boldsymbol{g} : \mathcal{X} \to \mathcal{Y}$ under the learning criterion satisfies

$$\boldsymbol{g}(\boldsymbol{x}) = \boldsymbol{g}^{\star}(\boldsymbol{x}) \quad \text{a.e.} \tag{2}$$

**DT via Distribution Matching.** The core technique for DT is to learn a transfer map (or translation function) $\boldsymbol{g} : \mathcal{X} \to \mathcal{Y}$ that matches the distribution of the source domain transformed by $\boldsymbol{g}$ and that of the target domain (Zhu et al., 2017; De Bortoli et al., 2024)—that is, solving the following learning criterion:

$$\text{find invertible } \boldsymbol{g}$$
$$\text{s.t. } [\boldsymbol{g}]_{\#} p_{\boldsymbol{x}} = p_{\boldsymbol{y}} \tag{3}$$

A variety of distribution-matching tools have been used to enforce (3), including adversarial losses based on genera-

tive adversarial networks (GANs) (Goodfellow et al., 2014) and integral probability metrics such as maximum mean discrepancy (MMD) (Gretton et al., 2012). Recent works, such as flow matching (Liu et al., 2023; Albergo et al., 2025) and schrodinger bridges (De Bortoli et al., 2024; Shi et al., 2023), also consider dynamic distribution transport perspective. Many methods further introduce regularizers to encourage invertibility (Zhu et al., 2017), structural consistency (Park et al., 2020), optimal transport (Kornilov et al., 2024; Bunne et al., 2023), or other application-based desirable properties (Xu et al., 2022; Xie et al., 2023). Nevertheless, explicit distribution matching remains the central ingredient across these approaches.

**MPAs and Non-identifiability.** Using DT as backbone of many domain transfer applications has enjoyed empirical success (see, e.g., (Zhu et al., 2017; Choi et al., 2020; Bunne et al., 2023)). However, it is well-known that it does not identify the ground-truth function $g^\star$ uniquely. A key obstruction to identifiability in DT is the existence of non-trivial *measure-preserving automorphisms* (MPAs), i.e., function $\neq$ Id that does not alter the input distribution (Moriakov et al., 2020; Shrestha & Fu, 2024):

**Definition 2.2** (Measure-preserving automorphism (MPA)). A continuous bijective function $m : \mathcal{X} \to \mathcal{X}$ is an MPA of $p_{\boldsymbol{x}}$ if $[\boldsymbol{m}]_{\#p_{\boldsymbol{x}}} = p_{\boldsymbol{x}}$.

If $\boldsymbol{m}$ is a non-trivial MPA of $p_{\boldsymbol{x}}$ (i.e., $\boldsymbol{m} \neq \text{Id}$), then $g^\star \circ \boldsymbol{m}$ also satisfies $[g^\star \circ \boldsymbol{m}]_{\#p_{\boldsymbol{x}}} = p_{\boldsymbol{y}}$. Hence Problem (3) could return $\boldsymbol{g} = g^\star \circ \boldsymbol{m}$ instead of $g^\star$.

The existence of MPAs result in serious practical ramifications (Moriakov et al., 2020; Galanti et al., 2018; Shrestha & Fu, 2024): translation using $\boldsymbol{g} = g^\star \circ \boldsymbol{m}$ could lead to *content misalignment* between the input and translated samples. An example is shown in Fig. 1, where the digit "7" is translated into a different rotated digit by various UDT methods based on distribution matching only. Fig. 2 further illustrates how MPA could result in content misalignment in the above example. For simplicity, consider that the MNIST digit distribution $p_{\boldsymbol{x}}$ were a Gaussian $\mathcal{N}(\mu, \sigma^2)$. Since $\boldsymbol{m} : x \mapsto -x + 2\mu$ is an MPA for the Gaussian $p_{\boldsymbol{x}}$, both $g^\star$ and $g^\star \circ \boldsymbol{m}$ could be a solution of Problem (3). However, $g^\star$ and $g^\star \circ \boldsymbol{m}$ output different rotated digits ("4" and "3", respectively) for a given input digit "4".

Even worse, the literature (Shrestha & Fu, 2024; Moriakov et al., 2020) has reported that MPA exists for all continuous distributions under mild conditions. This means that without further information or structural constraints, DT remains ill-posed and the $g^\star$ is non-identifiable.

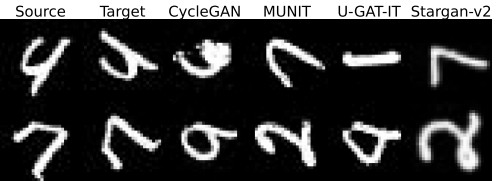

*Figure 1.* Domain 1: MNIST digits, Domain 2: rotated MNIST digits. MPA issue causes content-misaligned translation as depicted by the change in digit identity after translation by existing methods.

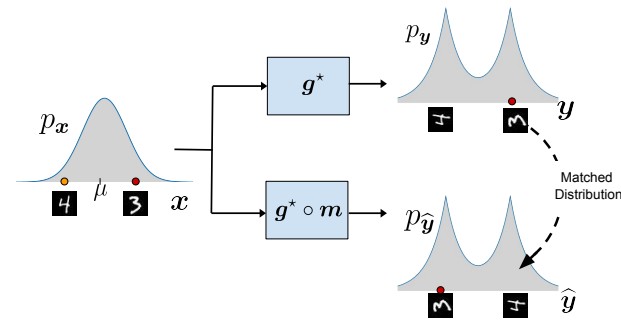

*Figure 2.* Illustration of how MPA results in content misalignment. If $p_{\boldsymbol{x}}$ were distributed $\mathcal{N}(\mu, \sigma^2)$, $\boldsymbol{m} = -x + 2\mu$ would be an MPA. Hence $g^\star \circ \boldsymbol{m}$ and $g^\star$ both are solutions to the distribution transport Problem (3). However, $g^\star \circ \boldsymbol{m}$ results in content misalignment.

### 2.2. Prior Art: Diversified Distribution Matching

To address the non-identifiability challenge brought by MPA, recent works (Shrestha & Fu, 2024; 2025) proposed to *diversify* distribution matching, i.e., align multiple distribution pairs rather than only the global marginals as in (3). In particular, they assume the availability of per-sample auxiliary information $u$ (e.g., a label, attribute, or side information) such that we can form cross-domain conditionals $\{p_{\boldsymbol{x}|u=u_i}\}_{i=1}^I$ and $\{p_{\boldsymbol{y}|u=u_i}\}_{i=1}^I$. They then enforce distribution matching for each conditional pair, i.e.,

> find invertible $\boldsymbol{g}$
> s.t. $[\boldsymbol{g}]_{\#p_{\boldsymbol{x}|u=u_i}} = p_{\boldsymbol{y}|u=u_i}, \qquad \forall i \in [I],$    (4)

The main insight established in (Shrestha & Fu, 2024) is that MPAs that simultaneously preserve *many* diverse conditionals become increasingly unlikely to exist—which assists establishing identifiability of $g^\star$.

The DDM identifiability theory provides a practical and effective route to MPA elimination. However, DDM has a major drawback: it requires the availability of auxiliary information for all samples. Specifically, each $\boldsymbol{x}$ and $\boldsymbol{y}$ must be annotated with the variable $u$ that identifies which conditional distributions $p_{\boldsymbol{x}|u}$ and $p_{\boldsymbol{y}|u}$ it is drawn from. This auxiliary information may be unavailable or costly to obtain in many applications. For example, in single-cell analysis (Tong et al., 2020; Saelens et al., 2019), it is

unclear what kind of side information could be used as $u$ across various phases of cell development.

## 2.3. Nonlinear Unmixing and DT

To reduce the amount of side information, our idea is to leverage a subtle but interesting connection between nonlinear unmixing and DT.

Nonlinear unmixing (e.g., (Hyvärinen & Pajunen, 1999; Hyvarinen et al., 2019; Khemakhem et al., 2020; Lachapelle et al., 2022; Zheng et al., 2022)) aims at recovering latent components $x$ from their observed nonlinear mixtures $y = g^\star(x)$, where $g^\star$ is an unknown invertible "mixing" function. There are many fundamental limitations of nonlinear unmixing. One of them is that $x$ is often only identifiable up to permutation and component-wise invertible transformations (Hyvärinen & Pajunen, 1999; Khemakhem et al., 2020; von Kügelgen, 2024). To be specific, let $\widehat{x}$ be the recovered sources. Then the individual element of $\widehat{x}$, denoted by $\widehat{x}_d$, is related to $x$ by an index permutation function $\pi : [D] \to [D]$ and a component-wise invertible transformation $h_d : \mathbb{R} \to \mathbb{R}, d \in [D]$:

$$\widehat{x}_d = h_d(x_{\pi(d)}), \tag{5}$$

where $d \in [D]$ is the index of the source component. In matrix notation, let $\mathbf{\Pi} \in \mathbb{R}^{D \times D}$ be a permutation matrix, then $\widehat{x} = h(\mathbf{\Pi}x)$, where $h(x) = [h_1(x_1), \ldots, h_D(x_D)]^T$. The fundamental limitations arise for a number of reasons; see (Von Kügelgen et al., 2021) for discussion.

If one views the latent components and observed data in nonlinear unmixing as the source domain and target domain in DT, respectively, then the two problems exhibit similar mathematical forms. Then, using any nonlinear unmixing approaches could identify $f(y) = h(\mathbf{\Pi}x)$, which is *almost* the inverse of $g^\star$. Nonetheless, the existence of $h$ and $\mathbf{\Pi}$ is not acceptable to DT, as they distort translated features. Nonetheless, as in DT one also observes $x \sim p_x$ (i.e., $x$ is not "latent" in DT), it renders opportunities to eliminate $h$ and $\mathbf{\Pi}$ on top of regular nonlinear unmixing—this is the starting point of our approach.

## 3. Nonlinear Unmixing-Assisted DT

We first introduce two assumptions used in our main theorem. The first assumption imposes the following structural condition on the Jacobian support pattern of the map $g^\star$.

**Assumption 3.1** (A3.1 (Structural sparsity)). For all $k \in \{1, \ldots, D\}$, there exists $\mathcal{C}_k \subseteq \{1, \ldots, D\}$ such that

$$\bigcap_{i \in \mathcal{C}_k} \mathcal{F}_{i,:} = \{k\}, \tag{6}$$

where $\mathcal{F} = \mathrm{supp}(\boldsymbol{J}_{g^\star}(\mathcal{X}))$ and $\boldsymbol{J}_{g^\star}(x)$ denotes the Jacobian matrix of $g^\star$ at location $x$.

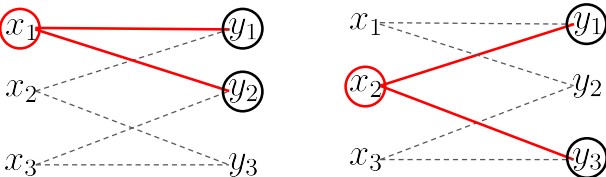

*Figure 3.* Illustration of Assumption 3.1 on the structure of $g^\star$. For each input $x_k$ (in red circle), one can find a subset of indices $\mathcal{C}_k$ (in black circles) corresponding to output coordinates that are influenced by $x_k$, such that $x_k$ is the only common influence (illustrated by solid red lines) for these outputs.

Assumption 3.1 follows the recent development in nonlinear unmixing via sparse structure (Zheng et al., 2022). It captures the idea that an input coordinate $x_i$ only influences upon a small subset of output coordinates. We argue that the assumption suits for high-dimensional DT problems as well, a source feature might only affect a subset of target features as many feature transformations are relatively "local" (e.g., in image-to-image translation). An example of such structure $g^\star$ satisfying Assumption 3.1 is shown in Figure 3.

Next, in addition to unpaired samples from $p_x$ and $p_y$, we assume access to at least one paired anchor sample.

**Assumption 3.2** (One paired anchor). There exists at least one paired sample $(x^{(\ell)}, y^{(\ell)})$ such that $x^{(\ell)} \sim p_x$ and $y^{(\ell)} = g^\star(x^{(\ell)})$.

This type of side information is arguably (much) easier to acquire compared to that in DDM (Shrestha & Fu, 2024; 2025), where each sample needs to be assigned to a conditional distribution.

## 3.1. Proposed DT Criterion

Under Assumptions 3.1–3.2, we propose to learn $g$ by the following criterion:

$$\min_{g} \quad \mathbb{E}_x \big\| \boldsymbol{J}_g(x) \big\|_0 \tag{7a}$$

$$\text{s.t.} \quad [g]_\# p_x = p_y, \tag{7b}$$

$$g(x^{(\ell)}) = y^{(\ell)}, \; \ell \in \mathcal{E} \tag{7c}$$

$$g \text{ is invertible}, \tag{7d}$$

where $\mathcal{E}$ is the index set of aligned sample pairs. Here $\| \cdot \|_0$ counts the number of nonzero entries.

Let $\widehat{g}$ be any solution to (7). Let $\widehat{\mathcal{F}} = \mathrm{supp}(\boldsymbol{J}_{\widehat{g}}(\mathcal{X}))$ and $\mathcal{V} = \mathrm{supp}(\{\boldsymbol{J}_{g^{-1} \circ \widehat{g}}(\mathcal{X})\})$. Let $\mathcal{R}_{\mathcal{B}}^D$ denote a subspace of $\mathbb{R}^D$ with all elements outside of $\mathcal{B} \subseteq [D]$ being zero. Then consider the following regularity assumption from (Zheng et al., 2022):

**Assumption 3.3.** Suppose that for all $d \in [D]$, there exist $\{x^{(\ell)}\}_{\ell=1}^{|\mathcal{F}_{d,:}|}$ and a matrix $\boldsymbol{V} \in \mathbb{R}^{D \times D}$ with $\mathrm{supp}(\boldsymbol{V}) =$

$\mathcal{V}$ such that $\mathrm{span}\left(\{J_{g^\star}(x^{(\ell)})_{d,:}\}_{\ell=1}^{|\mathcal{F}_{d,:}|}\right) = \mathbb{R}_{\mathcal{F}_{d,:}}^D$ and $[J_{g^\star}(x^{(\ell)})V]_{d,:} \in \mathbb{R}_{\mathcal{F}_{d,:}}^D$.

Assumption 3.3 is a commonly used assumption in sparse nonlinear unmixing literature (Zheng et al., 2022; Zheng & Zhang, 2023) to prevent pathological cases (e.g., all samples being limited to a degenerate subspace).

Using Problem (7), we state the following identifiability theorem:

**Theorem 3.4** (Identifiability via anchored nonlinear unmixing). *Consider the data model* (1). *Suppose that Assumptions 3.1, 3.2, and 3.3 hold, and that* $|\mathcal{E}| \geq 1$. *Then with probability one,*

$$\widehat{g} = g^\star \qquad a.e. \tag{8}$$

Theorem 3.4 establishes translation identifiability based on very light side information: instead of requiring labels or attributes for all samples as in DDM (Shrestha & Fu, 2024; 2025), we require as few as a single aligned sample. In practice, even under model mismatches and noise, using $1 < |\mathcal{E}| \leq 10$ for fairly high-dimensional data (e.g., $128 \times 128$ images) usually suffices to produce sensible results.

The complete proof of the theorem is provided in Appendix A. A brief sketch is as follows:

*Proof.* The proof consists of three steps.

*Step 1: Nonlinear unmixing yields a finite ambiguity class.* Under Assumption 3.1, Jacobian sparsity minimization of (7a) places $\widehat{g}$ in a nonlinear unmixing regime. Existing nonlinear unmixing result in (Zheng et al., 2022) implies that any minimizer must take the form

$$\widehat{g}^{-1}(y) = h(\Pi x), \text{ where } y = g^\star(x), \tag{9}$$

where $\Pi$ is a permutation matrix and $h$ is component-wise invertible. Thus, Jacobian sparsity alone narrows the solution space to a structured equivalence class.

*Step 2: Distribution matching turns the remaining ambiguity into MPAs.* The constraint (7b) enforces $\widehat{y} = \widehat{g}(x) \overset{d}{=} y$. Combined with (9), this implies that each coordinate mapping $h_i$ must be a one-dimensional translation from $x_{\pi(i)}$ to $x_i$, i.e.,

$$x \overset{d}{=} \widehat{g}^{-1}(y) \implies x \overset{d}{=} h(\Pi x)$$
$$\implies x_i \overset{d}{=} h_i(x_{\pi(i)}), \quad i \in [D].$$

Importantly, we show that the admissible choices of $h_i$ are limited (only a finite number of admissible functions that can satisfy $x_i \overset{d}{=} h_i(x_{\pi(i)})$).

*Step 3: A single anchor eliminates permutations and non-trivial MPAs almost surely.* Finally, the anchor constraint

(7c) requires $\widehat{g}(x^{(\ell)}) = y^{(\ell)}$, i.e.,

$$x^{(\ell)} = h(\Pi x^{(\ell)}). \tag{10}$$

We show that for any non-trivial coordinate-wise MPA, the set of fixed points is a singleton, and for any nontrivial permutation $\Pi \neq I$, the set of $x$ satisfying $h(\Pi x) = x$ has measure zero under $p_x$. Therefore, a randomly drawn anchor $x^{(\ell)}$ rules out all $(\Pi, h)$ except $(I, \mathrm{Id})$ with probability one, yielding $\widehat{g} = g^\star$ a.e. □

# 4. Scalable Implementation

## 4.1. GAN-based learning objective for (7)

We implement (7) by parameterizing $g$ via a neural network $g_\theta : \mathcal{X} \to \mathcal{Y}$, and optimizing it via adversarial distribution matching (Goodfellow et al., 2014), an anchor loss, and a carefully designed differentiable surrogate for the Jacobian sparsity objective. Concretely, let $d_\psi$ denote the discriminator. Then, we optimize the following objective:

$$\min_\theta \max_\psi \mathcal{L}_{\mathrm{GAN}}(g_\theta, d_\psi) + \lambda_{\mathrm{anch}}\mathcal{L}_{\mathrm{anch}}(g_\theta) \tag{11}$$
$$+ \lambda_{\mathrm{sp}}\mathcal{L}_{\mathrm{sp}}(g_\theta) + \lambda_{\mathrm{inv}}\mathcal{L}_{\mathrm{inv}}(g_\theta),$$

where $\mathcal{L}_{\mathrm{GAN}}, \mathcal{L}_{\mathrm{anch}}$, and $\mathcal{L}_{\mathrm{inv}}$ enforces the constraints (7b), (7c), and (7d), respectively, and $\mathcal{L}_{\mathrm{sp}}$ corresponds to the objective (7a).

Here, $\mathcal{L}_{\mathrm{GAN}}$ is the standard GAN distribution matching loss (Goodfellow et al., 2014):

$$\mathcal{L}_{\mathrm{GAN}}(g_\theta, d_\psi) = \tag{12}$$
$$\mathbb{E}_{y \sim p_y}[\log d_\psi(y)] + \mathbb{E}_{x \sim p_x}[\log(1 - d_\psi(g_\theta(x)))].$$

The anchor loss $\mathcal{L}_{\mathrm{anch}}$ is a simple regression form

$$\mathcal{L}_{\mathrm{anch}}(g_\theta) = \left\| g_\theta(x^{(\ell)}) - y^{(\ell)} \right\|_2^2. \tag{13}$$

The $\mathcal{L}_{\mathrm{inv}}(g_\theta)$ term is an invertibility promoting regularization realized as follows:

$$\mathcal{L}_{\mathrm{inv}}(g_\theta) = \min_\alpha \ \mathbb{E}_{x \sim p_x}\|f_\alpha(g_\theta(x)) - x\|_1 \tag{14}$$

Here, $f_\alpha$ is another neural network that is optimized to reconstruct $x$ from $g_\theta(x)$. Under sufficiently expressive $f_\alpha$, $\mathcal{L}_{\mathrm{inv}}$ attains the minimum zero only when $g_\theta$ is invertible, i.e., at $f_\alpha = g_\theta^{-1}$; see (Zhu et al., 2017) for similar designs.

Lastly, the $\mathcal{L}_{\mathrm{sp}}(g_\theta)$ term is a computable surrogate that approximates $\mathbb{E}_x\|J_{g_\theta}(x)\|_0$ which will be discussed in the next subsection.

## 4.2. Finite-difference based Jacobian sparsity regularization

The Jacobian $J_{g_\theta}(x) \in \mathbb{R}^{D \times D}$ is high-dimensional for large $D$. Typically, extracting the full Jacobian for a

neural network in general requires *one backpropagation per row/column*, i.e., $O(D)$ backward passes per sample—which quickly becomes the computational bottleneck. Moreover, storing the jacobian requires $O(D^2)$ space, which can also limit the batch size during training.

**Masked Finite-difference surrogate.** When $D$ is small, it is possible to compute and store $\boldsymbol{J}$ using $D$ differentiations. Hence, we can use

$$\mathcal{L}_{\mathrm{sp}}(\boldsymbol{g}_{\boldsymbol{\theta}}) = \mathbb{E}_{\boldsymbol{x}} \| \boldsymbol{J}_{\boldsymbol{g}_{\boldsymbol{\theta}}}(\boldsymbol{x}) \|_1, \tag{15}$$

to approximate the sparsity of the Jacobian. However for large $D$, computing jacobian quickly becomes the bottleneck during training. One way to avoid multiple backward passes, which are computationally more expensive than forward passes, is to use finite-difference based approximation. Consider a one-hot mask $\boldsymbol{e}_d = [0, \ldots, 0, 1, 0, \ldots, 0]$ with 1 at the $d$-th position and 0 elsewhere. Then, the $d$-th column of the Jacobian is given by $\boldsymbol{J}_{\boldsymbol{g}_{\boldsymbol{\theta}}}(\boldsymbol{x})_{:,d} = \frac{\boldsymbol{g}_{\boldsymbol{\theta}}(\boldsymbol{x}+\delta \boldsymbol{e}_d)-\boldsymbol{g}_{\boldsymbol{\theta}}(\boldsymbol{x})}{\delta}$. Then,

$$\left\| \boldsymbol{J}_{\boldsymbol{g}_{\boldsymbol{\theta}}}(\boldsymbol{x}) \boldsymbol{e}_d \right\|_0 \approx \left\| \frac{\boldsymbol{g}_{\boldsymbol{\theta}}(\boldsymbol{x} + \delta \boldsymbol{e}_d) - \boldsymbol{g}_{\boldsymbol{\theta}}(\boldsymbol{x})}{\delta} \right\|_0. \tag{16}$$

Hence, $\left\| \boldsymbol{J}_{\boldsymbol{g}_{\boldsymbol{\theta}}}(\boldsymbol{x}) \right\|_0 \approx \sum_{d=1}^{D} \left\| \boldsymbol{J}_{\boldsymbol{g}_{\boldsymbol{\theta}}}(\boldsymbol{x}) \boldsymbol{e}_d \right\|_0$. Nonetheless, this approach also requires $D$ forward passes, which is still computationally expensive for large $D$. The issue is that perturbation to $\boldsymbol{x}$ inside $\boldsymbol{g}_{\boldsymbol{\theta}}(\boldsymbol{x} + \delta \boldsymbol{e}_d)$ is applied one dimension at a time, which requires $D$ forward passes to evaluate the Jacobian sparsity, raising complexity concerns.

To avoid repeated forward passes, we explore an alternative in which we apply the perturbation to multiple dimensions at once, i.e., consider a random mask $\boldsymbol{z} \in \mathbb{R}^D$, such that $\|\boldsymbol{z}\|_0 = S \ll D$. Now consider the following alternative finite difference approximation:

$$\left\| \boldsymbol{J}_{\boldsymbol{g}_{\boldsymbol{\theta}}}(\boldsymbol{x}) \boldsymbol{z} \right\|_0 \approx \left\| \frac{\boldsymbol{g}_{\boldsymbol{\theta}}(\boldsymbol{x} + \delta \boldsymbol{z}) - \boldsymbol{g}_{\boldsymbol{\theta}}(\boldsymbol{x})}{\delta} \right\|_0. \tag{17}$$

If we minimize $\|\boldsymbol{J}_{\boldsymbol{g}_{\boldsymbol{\theta}}}(\boldsymbol{x}) \boldsymbol{z}\|_0$ instead of $\|\boldsymbol{J}_{\boldsymbol{g}_{\boldsymbol{\theta}}}(\boldsymbol{x}) \boldsymbol{e}_d\|_0$, we are enforcing sparse changes in the input to $\boldsymbol{g}_{\boldsymbol{\theta}}$ to cause sparse changes to its output. This is intuitively linked to the Jacobian sparsity.

Interestingly, we can also show that this masked finite-difference is a provable proxy for the Jacobian sparsity in upper/lower bounding sense. We consider a sparse Gaussian probe, i.e., $z_i \sim \mathcal{N}(0, 1)$ when $z_i \neq 0$, as the perturbation. In practice, other continuous distributions can also be used for $z_i$.

**Proposition 4.1.** *Let* $\boldsymbol{J} \in \mathbb{R}^{D \times D}$. *For each row* $d \in \{1, \ldots, D\}$, *define the support* $\mathcal{T}_d := \{j \in \{1, \ldots, D\} : \boldsymbol{J}_{dj} \neq 0\}$ *and its size* $T_d := |\mathcal{T}_d|$ *and* $T = \max_d T_d$. *Fix an integer* $1 \leq S \leq D$, *and draw a random probe* $\boldsymbol{r} \sim$

$\mathrm{Unif}(\mathcal{R}_{D,S})$, *where* $\mathcal{R}_{D,S} = \{\boldsymbol{r} \in \{0,1\}^D \mid \|\boldsymbol{r}\|_0 = S\}$ *is the set of all masks with exactly* $S$ *elements* $= 1$ *and others* 0. *Let* $\boldsymbol{z} = \boldsymbol{r} \odot \boldsymbol{\epsilon}$, *where* $\boldsymbol{\epsilon} \sim \mathcal{N}(\boldsymbol{0}, \boldsymbol{I}_D)$. *Define*

$$q(\boldsymbol{J}) = \frac{D}{S} \mathbb{E}_{\boldsymbol{z}} \| \boldsymbol{J} \boldsymbol{z} \|_0 \tag{18}$$

*Then* $q(\boldsymbol{J})$ *satisfies the following bound*

$$\|\boldsymbol{J}\|_0 \geq q(\boldsymbol{J}) \geq \left(1 - \frac{(S-1)(T-1)}{2(D-1)}\right) \|\boldsymbol{J}\|_0. \tag{19}$$

The proof is given in Appendix B. Proposition 4.1 shows that $q(\boldsymbol{J})$ closely approximates $\|\boldsymbol{J}\|_0$ when $(S - 1)(T - 1) \ll D$. Hence, selecting sparse enough random probe $\boldsymbol{z}$ for sufficiently sparse high-dimensional Jacobian $\boldsymbol{J}$ allows us to approximate $\|\boldsymbol{J}\|_0$ using $q(\boldsymbol{J})$ with high accuracy. Note that when $S = 1$, $\|\boldsymbol{J}\boldsymbol{z}\|_0 = \|\boldsymbol{J}_{:,d}\|_0$ for some $d \in [D]$. This is equivalent to the standard finite-difference approximation which requires $\mathcal{O}(D)$ number of $\boldsymbol{z}$ samples for accurate approximation. However, when $S > 1$, we need fewer $\boldsymbol{z}$ samples to approximate $\|\boldsymbol{J}\|_0$. This is because $\boldsymbol{z}$ considers multiple columns of $\boldsymbol{J}$ at once resulting in lower variance estimation than single column-wise estimation (see Appendix E.5 for more details).

Therefore for large $D$, we use the following finite difference $\ell_1$ approximation of $q(\boldsymbol{J})$ as the sparsity regularizer:

$$\mathcal{L}_{\mathrm{sp}}(\boldsymbol{g}_{\boldsymbol{\theta}}) = \mathbb{E}_{\boldsymbol{x},\boldsymbol{z}} \left\| \frac{\boldsymbol{g}_{\boldsymbol{\theta}}(\boldsymbol{x} + \delta \boldsymbol{z}) - \boldsymbol{g}_{\boldsymbol{\theta}}(\boldsymbol{x})}{\delta} \right\|_1 \tag{20}$$

Note that the constant $D/S$ can be absorbed by $\lambda_{\mathrm{sp}}$.

# 5. Numerical Experiments

We validate our theoretical claims on 2D synthetic data and image-to-image translation (MNIST to rotated MNIST and Edges to rotated shoes) tasks.

## 5.1. 2D synthetic data

We generate samples from the two domains $\boldsymbol{x} \in \mathbb{R}^2$ and $\boldsymbol{y} \in \mathbb{R}^2$ as follows:

$$y_1 \sim \mathrm{Unif}[-1, 1], \quad y_2 \sim \mathrm{Unif}[-1, 1] + 0.5 y_1$$
$$\boldsymbol{x} = t \cos(\boldsymbol{A}\boldsymbol{y}) + \boldsymbol{A}\boldsymbol{y}, \tag{21}$$

where $\boldsymbol{A} \in \mathbb{R}^{2 \times 2}$ is a permutation matrix other than the identity, and $t \sim \mathrm{Unif}[0.3, 0.5]$ is a constant scaling factor. Here $\boldsymbol{y} \mapsto \boldsymbol{x}$ in (21) denotes the $\boldsymbol{g}^{\star -1}$ map.

**Metrics:** We use the translation error (TE) to measure the similarity between target and translated samples. Per-sample TE is defined as follows:

$$TE = \frac{1}{\sqrt{D}} \|\widehat{\boldsymbol{g}}_{\boldsymbol{\theta}}(\boldsymbol{x}) - \boldsymbol{y}\|_2, \tag{22}$$

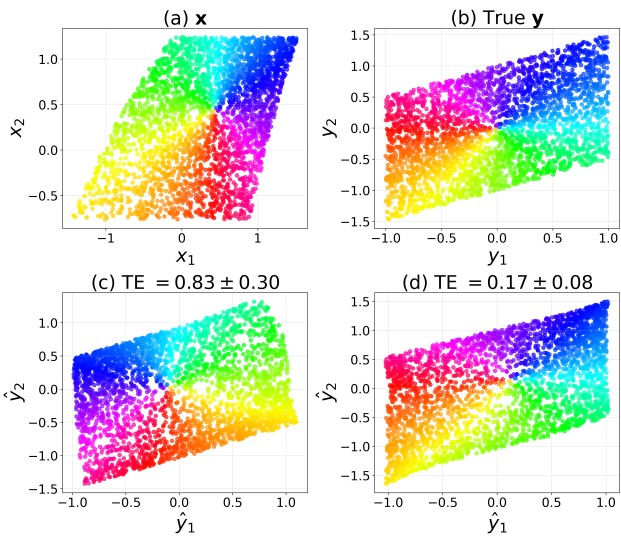

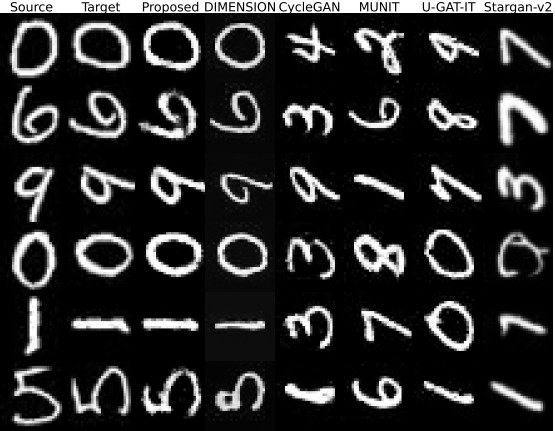

*Figure 5.* Result of translation from MNIST digits to rotated MNIST digits for all methods.

*Figure 4.* Scatter plot of (a) source samples (b) target samples, (c) translated samples when $\lambda_{\text{anch}} = 0$, and (d) translated samples when $\lambda_{\text{anch}} = 1$. Corresponding samples $\boldsymbol{x}$, $\boldsymbol{y} = \boldsymbol{g}^\star(\boldsymbol{x})$ and $\widehat{\boldsymbol{y}} = \widehat{\boldsymbol{g}}(\boldsymbol{x})$ are coded with the same color.

where $\boldsymbol{x}$ and $\boldsymbol{y}$ are the source and target samples, respectively. We report the mean and standard deviation of TE over all samples.

**Setting:** We use a 2-layer MLP with 32 hidden dimensions and leaky ReLU activation to represent $\boldsymbol{g}_{\boldsymbol{\theta}}$. We use 1 anchor sample with $\lambda_{\text{sp}} = 0.1$, and $\lambda_{\text{inv}} = 1.0$. Since it is a low dimensional setting, we use Eq.(15) as $\mathcal{L}_{\text{sp}}$. Additional settings are provided in the appendix.

**Results:** Fig. 4 shows the scatter plots of the samples from (a) $\boldsymbol{x}$, (b) $\boldsymbol{y}$, (c) $\widehat{\boldsymbol{g}}_{\boldsymbol{\theta}}(\boldsymbol{x})$ when $\lambda_{\text{anch}} = 0$ (i.e., without any anchor paired sample), and (d) $\widehat{\boldsymbol{g}}_{\boldsymbol{\theta}}(\boldsymbol{x})$ when $\lambda_{\text{anch}} = 1$. The samples are color coded such that $\boldsymbol{x}$, $\boldsymbol{y} = \boldsymbol{g}^\star(\boldsymbol{x})$, and $\widehat{\boldsymbol{y}} = \widehat{\boldsymbol{g}}(\boldsymbol{x})$ share the same color. For the translation to be correct, we expect $\widehat{\boldsymbol{y}} = \boldsymbol{y}$. Nonetheless, in Fig. 4 (b) and (c), we observe that the samples are not aligned with the target samples $\boldsymbol{y}$. The TE of 0.83 is also quite high. This indicates that the translation is not correct although the supports of $\widehat{\boldsymbol{y}}$ and $\boldsymbol{y}$ are matched due to successful distribution matching. But after using a single anchor in Fig. 4 (d), we observe that the samples are aligned with the target samples $\boldsymbol{y}$, and the TE is reduced to 0.17.

### 5.2. Image-to-image translation

We evaluate the proposed method on challenging image to image translation tasks. Since these are high dimensional settings, we use masked finite-difference in Eq. (20) for the sparsity regularizer.

Note that the structural sparsity Assumption 3.1 also has minor architectural implications, particularly for normalization

layers: standard instance normalization couples all spatial locations within a channel and thus precludes a sparse Jacobian. We therefore replace instance normalization with channel-only normalization in our image experiments. We did not observe any degradation from this replacement, and most other architectural choices remain compatible.

**Baselines:** We compare the proposed method with various DT methods: `DIMENSION` (Shrestha & Fu, 2024), `CUT` (Park et al., 2020), `CycleGAN` (Zhu et al., 2017), `U-GAT-IT` (Kim et al., 2020), `UNIT` (Liu et al., 2017), `MUNIT` (Huang et al., 2018), `StarGAN-v2` (Choi et al., 2020), `Hneg-SRC` (Jung et al., 2022), `GP-UNIT` (Yang et al., 2023), `OverLORD` (Gabbay & Hoshen, 2021), `ZeroDIM` (Gabbay et al., 2021).

### 5.2.1. MNIST DIGITS TO ROTATED MNIST DIGITS

We use MNIST digits as the first domain $p_{\boldsymbol{x}}$ and 90 degrees rotated MNIST digits as the second domain $p_{\boldsymbol{y}}$. It is clear that the translation simply needs to rotate the image by 90 degrees. However, this task is challenging for methods based only on distribution matching because of the existence of MPA, which results in digit identity-misaligned translation.

**Settings:** We use a randomly sampled single anchor pair for the proposed method. We use $\lambda_{\text{sp}} = 0.01$, and $\lambda_{\text{anch}} = 1.0$. Additional settings and baselines are provided in the appendix.

**Results:** Fig. 5 shows the result of translation for all methods. We observe that the proposed method and `DIMENSION` (Shrestha & Fu, 2024) successfully fix the content misalignment issue. Note that `DIMENSION` uses digit label as the side information to diversify distribution matching. Arguably, a single anchor sample is more easily available than labels for all samples in practice. Table 1

*Table 1.* Quantitative alignment results for image translation tasks. "E", "rS", "M", and "rM" denote Edges, rotated Shoes, MNIST, and rotated MNIST, respectively.

| Method | LPIPS($\downarrow$) | TE($\downarrow$) |
|---|---|---|
| | $E \to rS$ | $M \to rM$ |
| Proposed | $0.32 \pm 0.07$ | $\mathbf{0.20 \pm 0.06}$ |
| DIMENSION | $\mathbf{0.29 \pm 0.06}$ | $0.35 \pm 0.09$ |
| CUT | $0.65 \pm 0.10$ | $0.81 \pm 0.19$ |
| CycleGAN | $0.65 \pm 0.03$ | $0.60 \pm 0.10$ |
| U-GAT-IT | $0.56 \pm 0.05$ | $0.62 \pm 0.11$ |
| UNIT | $0.49 \pm 0.03$ | $0.65 \pm 0.10$ |
| MUNIT | $0.50 \pm 0.03$ | $0.64 \pm 0.10$ |
| StarGAN-v2 | $0.39 \pm 0.05$ | $0.66 \pm 0.11$ |
| Hneg-SRC | $0.45 \pm 0.06$ | — |
| GP-UNIT | $0.49 \pm 0.08$ | — |
| OverLORD | $0.43 \pm 0.06$ | — |
| ZeroDIM | $0.38 \pm 0.06$ | — |

shows the TE attained by various methods for this task. One can see that the proposed method achieves the lowest TE among all methods.

### 5.2.2. EDGES TO ROTATED SHOES

We use the popular image translation dataset Edges vs Shoes. However, to make the task more challenging, we intentionally rotate the shoes by 90 degrees as in (Shrestha & Fu, 2024). This simple rotation makes the content-misalignment issue appear more prominently in existing methods.

**Setting:** For complex high-dimensional distributions like images, we observed that we require more than one anchor sample in practice. For this more complex task, we use 10 anchor samples for the proposed method. This gap from the single-anchor sufficiency in Theorem 3.4 can be attributed to two facts: (i) the theoretical assumptions may not be exactly met by complex real-world distributions (e.g., the Jacobian is only approximately sparse), and (ii) the nonconvex min-max problem in Eq. (11) cannot be solved exactly/optimally in practice. Adding a few extra anchors compensates for these mismatches without altering the spirit of the theory. Note that DIMENSION is a DDM-based approach that requires supervision. Here, DIMENSION used shoe type (sandal, boot, shoe, heels) as the side information. Additional settings are described in the appendix.

**Results:** Table 1 shows the LPIPS attained by various methods for this task. LPIPS measures the perceptual similarity between the translated and target images using pretrained vision backbone. One can see that the proposed method achieves competitive LPIPS compared to DIMENSION. But the latter used substantially more supervision. An ablation on the number of anchors $|\mathcal{E}| \in \{1, 2, 5, 10\}$ for this task is provided in Appendix E.3 (Table 6).

### 5.3. Single-Cell Sequence Alignment

To further validate the proposed framework on a real-world scientific application, we consider single-cell sequence alignment, a core task in computational biology and AI4Science (Yang et al., 2021; Eyring et al., 2024; Amodio & Krishnaswamy, 2018). Single-cell measurement processes are destructive, so paired measurements across modalities (e.g., RNA-seq and ATAC-seq) on the same cell are difficult to obtain. The goal is therefore to translate between modalities using mostly unpaired data.

**Setting:** We focus on translation between ATAC-seq and RNA-seq measurements, using human lung adenocarcinoma A549 cells from (Cao et al., 2018). The dataset contains 1,874 samples, split into 1,534 training and 340 testing samples following (Yang et al., 2021). We apply TF-IDF preprocessing on the count data (fitted on the training set). Following (Yang et al., 2021), we use K-NN accuracy between translated and target samples as the evaluation metric. The translation function is a single linear layer trained for 150 epochs with batch size 64 and learning rate 0.002; remaining settings follow (Yang et al., 2021).

**Baseline:** We compare against the cross-modal autoencoder (CM-AE) baseline of (Yang et al., 2021).

**Results:** Figure 7 reports K-NN accuracy of RNA-seq to ATAC-seq translation. The proposed method substantially outperforms CM-AE across all $k$. Importantly, a single paired anchor only translates into a meaningful gain when combined with the Jacobian sparsity regularizer, mirroring the theoretical insight that anchor and sparsity together drive identifiability.

### 5.4. Ablation Studies

Ablation study for analysing the effect of the various loss terms (e.g., Jacobian sparsity regularization, anchor matching regularization, and distribution matching) is provided in Appendix E.2. Similarly, we also provide an ablation study on the efficacy of the proposed Jacobian sparsity regularizer in Appendix E.5.

For reproducibility, the source code is publicly available at: https://github.com/shresthasagar/Identifiable_DT_anchor.git

## 6. Conclusion

This work revisits the ill-posed nature of domain transfer. The core difficulty stems from the existence of measure-preserving automorphisms (MPAs)—mappings that leave a distribution unchanged—yet can arbitrarily alter cross-domain correspondences, leading to content-misaligned translation in practice. Prior approaches to eliminating MPAs typically require sample-wise side information (e.g.,

| Source | Target | Proposed | Dimension | CycleGAN | U-GAT-IT | Stargan-v2 | Hneg-SRC | GP-UNIT | ZeroDIM | CUT |
|--------|--------|----------|-----------|----------|----------|------------|----------|---------|---------|-----|

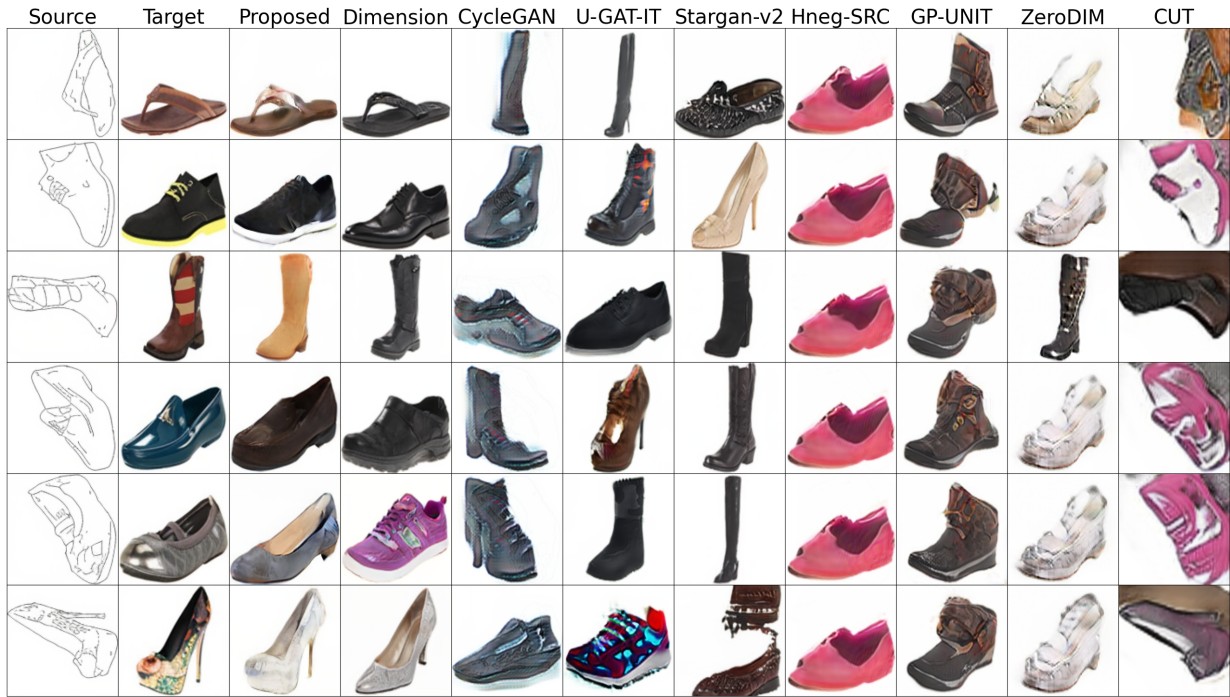

*Figure 6.* Result of translation from Edges to rotated shoes for all methods.

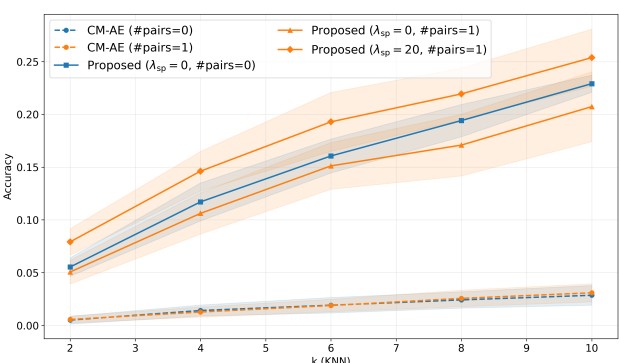

*Figure 7.* K-NN accuracy of RNA-seq to ATAC-seq translation on the A549 single-cell dataset (Cao et al., 2018), comparing the proposed method to CM-AE (Yang et al., 2021). The proposed method clearly benefits from anchor matching combined with sparsity regularization.

labels or attributes) for all samples. We propose an alternative route based on structural sparsity: under a Jacobian support sparsity condition, a single paired anchor sample suffices to rule out MPAs. This substantially relaxes the supervision requirements of existing identifiability results while achieving comparable alignment performance. For practical high-dimensional learning, we further introduce a Jacobian sparsity regularizer based on masked finite differences, making the approach applicable to data such as images. Experimental results corroborate the theory.

**Limitations.** First, our approach assumes access to paired anchor samples; in some settings, even a small number of anchors may be unavailable. An important direction is identifiability without any paired data. Second, the sparsity assumption may not hold for all domain transfer tasks; exploring alternative conditions that enable identifiability is a promising avenue for future work. Third, our analysis covers exact-pair anchors and the population case, but does not characterize robustness to noisy/approximate anchors or finite-sample effects. Establishing identifiability under practical settings generally requires different proof techniques (e.g., (Hyvarinen et al., 2019; Lyu & Fu, 2022; Lyu et al., 2022)) and constitutes an important line of future work.

## Acknowledgements

This work was supported in part by the National Science Foundation (NSF) under Project ECCS-2450987, and in part by the NSF CAREER Award ECCS-2144889.

## Impact Statement

This work focuses on theoretical foundations for multi-domain distribution transfer. It does not introduce new datasets, deployable systems, or application-specific technologies. We therefore do not anticipate any direct societal impacts, either positive or negative, beyond advancing the understanding of learning theory.

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

# Supplementary Materials of "Domain Transfer Becomes Identifiable via a Single Alignment"

## A. Proof of Theorem 3.4

> **Theorem 3.4** Consider the data model (1). Suppose Assumptions 3.1, 3.2, and 3.3 hold. Let $\widehat{g}$ be any solution to (7). Then with probability one,
> $$\widehat{g} = g^\star \qquad \text{a.e.}$$

*Proof.* We prove this in the following steps:

1. **Nonlinear Unmixing**: Note that $g^\star$ is a solution to Problem (7) with $h = g^\star$. Hence $\|J_{\widehat{g}}\|_0 \leq \|J_{g^\star}\|_0$. This fact together with Assumption 3.1 and 3.3 satisfies the conditions of Theorem 1 of (Zheng et al., 2022). Hence, we can conclude that
$$\widehat{g}^{-1}(y) = h(\Pi x) \tag{23}$$
where $h$ is component-wise invertible and $\Pi$ is a permutation matrix.

2. **Distribution Matching**: The constraint $\widehat{g}$ satisfies $[\widehat{g}]_{\#p_x} = p_y$. This implies that $p_x = [\widehat{g}^{-1}]_{\#p_y}$. Specifically, each component of $\widehat{x} := \widehat{g}^{-1}(y)$ has the same distribution as the corresponding component of $x$, i.e., $\widehat{x}_i = h_i(x_{\pi(i)})$.

We show that when the constraints (7b) and (7c) are satisfied, $\Pi = I$ and each $h = \text{Id}$, where $\text{Id}(\cdot)$ is the identity function.

The proof is based on the following lemmas, whose proofs will be provided in the following section.

**Lemma A.1** (One Unchanging Point). *Let $m$ be a smooth one-dimensional MPA of $p_x$, which is not the identity function. Then there does not exist more than one point $x$ such that $m(x) = x$.*

Based on lemma A.4, for a given permutation matrix $\Pi$, let us define $\mathcal{H}_\Pi$ as the set of all componentwise invertible functions $h$ such that $h(\Pi x) \stackrel{d}{=} x$. Note that $\mathcal{H}_\Pi$ is a finite set. Specifically, $|\mathcal{H}_\Pi| = 2^D$.

Also, define $\mathcal{R}$ as the set of all permutation matrices. Note that $|\mathcal{R}| = D!$.

**Lemma A.2** (Zero Measure Set). *The set $\mathcal{Z} = \{x \in \mathcal{X} \mid h(\Pi x) = x, h \in \mathcal{H}_\Pi, \Pi \in \mathcal{R}, \Pi \neq I\}$ has measure zero under $p_x$.*

Using Lemma A.2, we can conclude that $x^\ell \in \mathcal{Z}$ with probability zero under $p_x$. Therefore, $\Pi = I$. Now, it remains to show that $h = I$.

We have that $\widehat{x} = h(x) \stackrel{d}{=} x$, where $h$ is a component-wise invertible function. This implies that $h_i, \forall i \in [D]$ are component-wise MPA of $p_{x_i}$, since $h(x_i) \stackrel{d}{=} x_i$. We want to show that $h_i = \text{Id}, \forall i \in [D]$. We will show this by proving that if $h_i \neq \text{Id}$, then $h(x^\ell) = x^\ell$ cannot be satisfied.

The reason is that the set of points $x$ for which $h(x) = x$ is of measure zero if $h \neq \text{Id}$. To see this, recall that Lemma A.1 implies that the only one point remains fixed ($h_i(x) = x$) with respect to $h_i$ for all $i$ if $h_i$ is a non-trivial MPA. Therefore, if at least one $h_i$ is non-trivial (not identity), then the set $\mathcal{Z} = \{x \in \mathcal{X} \mid h(x) = x, \exists i \in [D] \text{ s.t. } h_i \neq \text{Id}\}$ has measure zero. This means that $h(x^\ell) = x^\ell$ with probability zero for randomly sampled $x^\ell$ if $h \neq \text{Id}$.

$\square$

### A.1. Proofs of Lemmas

First, we consider the following additional lemmas that are used in the proof of the main lemmas A.1 and A.2.

**Lemma A.3** (Uniqueness of MPA for one-dimensional r.v.). *There exists at most one non-identity differentiable function $m$ such that $[m]_{\#p_x} = p_x$ for one-dimensional continuous PDF $p_x$.*

**Lemma A.4** (Finite Translations). *Given any two one-dimensional distributions $p_1$ and $p_2$, there exists at most two non-identity smooth functions $r_1$ and $r_2$ such that $[r_1]_{\#p_1} = [r_2]_{\#p_1} = p_2$.*

Now, we provide the proofs for all the lemmas.

> **Lemma A.3** (Uniqueness of MPA for one-dimensional r.v.). There exists at most one non-identity differentiable function $m$ such that $[m]_{\#p_x} = p_x$ for one-dimensional continuous PDF $p_x$.

*Proof.* To show that any distribution $p_x$ has at most one non-trivial differentiable MPA, we use the fact that the uniform distribution $p_u = \mathcal{U}([0,1])$ has at most one non-trivial differentiable MPA.

Let $f_x$ denote the cumulative distribution function of $p_x$. Then, $[f_x]_{\#p_x} = p_u$.

To show that the differentiable MPA of $p_x$ is unique, suppose, for the sake of contradiction, that $m_1$ and $m_2$ are two distinct non-identity differentiable MPAs of $p_x$.

Then, we have:

$$[m_1]_{\#p_x} = [m_2]_{\#p_x} = p_x$$
$$[m_1 \circ f_x^{-1}]_{\#p_u} = [m_2 \circ f_x^{-1}]_{\#p_u} = p_x$$
$$[f_x \circ m_1 \circ f_x^{-1}]_{\#p_u} = [f_x \circ m_2 \circ f_x^{-1}]_{\#p_u} = p_u. \tag{24}$$

This implies that $f_x \circ m_1 \circ f_x^{-1}$ and $f_x \circ m_2 \circ f_x^{-1}$ are two distinct non-identity differentiable MPAs of $p_u$.

We show that $m_u : u \mapsto -u + 1$ and Id are the only differentiable MPAs of $p_u$. Suppose, for contradiction, that there exists another differentiable MPA $q$ of $p_u$, distinct from $m_u$ and Id. Then there exists $u' \in [0,1]$ such that $\frac{dq(u')}{du} \neq \pm 1$. Let $y = q(u), u \sim p_u$. By the change of variables formula for probability density functions,

$$p_u(u') = p_y(q(u')) \left| \frac{dq(u')}{du} \right|.$$

If $\left| \frac{dq(u')}{du} \right| \neq 1$, then $p_y(q(u')) \neq 1$. Hence $p_y \neq \mathcal{U}([0,1])$, contradicting $[q]_{\#p_u} = p_u$. Therefore the only differentiable MPAs of $p_u$ with $\frac{d}{du} q(u) \in \{-1, +1\}$ for all $u \in [0,1]$ are $m_u$ and Id, due to the boundary conditions $q(0) \in \{0,1\}$ and $q(1) \in \{0,1\}$ that any continuous MPA on $[0,1]$ must satisfy.

This contradicts (24). Therefore, $p_x$ has at most one non-identity differentiable MPA. $\qquad\square$

> **Lemma A.1** (One Unchanging Point). Let $m$ be a smooth one-dimensional MPA of $p_x$, which is not the identity function. Then there does not exist more than one point $x$ such that $m(x) = x$.

*Proof.* Assume that there exist two distinct points $x_1$ and $x_2$ such that $m(x_1) = x_1$ and $m(x_2) = x_2$.

Let $f_x$ be the cumulative distribution function of $p_x$. Then, from Lemma A.3, we know that $m_u = f_x \circ m \circ f_x^{-1}$ is the corresponding unique MPA of $p_u$.

However, since $m_u = u \mapsto -u + 1$, the only point that remains fixed with respect to $m_u$ is $\frac{1}{2}$. Therefore for $f_x(x_1) \in [0,1]$ and $f_x(x_2) \in [0,1]$, if $m_u(f_x(x_1)) = f_x(x_1)$, then $m_u(f_x(x_2)) \neq f_x(x_2)$. This implies that

$$m_u \circ f_x(x_2) \neq f_x(x_2)$$
$$\implies f_x \circ m \circ f_x^{-1} \circ f_x(x_2) \neq f_x(x_2)$$
$$\implies f_x \circ m(x_2) \neq f_x(x_2)$$
$$\implies m(x_2) \neq x_2$$

This is a contradiction. Therefore, there cannot be more than one point $x$ such that $m(x) = x$. $\qquad\square$

> **Lemma A.4** (Finite Translations). Given any two one-dimensional distributions $p_1$ and $p_2$, there exists at most two non-identity smooth functions $r_1$ and $r_2$ such that $[r_1]_{\#p_1} = [r_2]_{\#p_1} = p_2$.

*Proof.* Let there exist a pair of one-dimensional distributions $p_1$ and $p_2$ such that there are more than two non-identity smooth functions, i.e.,

$$\exists r_1, r_2, r_3, \text{ such that } r_i \neq r_j \forall i \neq j \text{ and } [r_1]_{\#} p_1 = [r_2]_{\#} p_1 = [r_3]_{\#} p_1 = p_2$$

Then one can construct two MPAs $m_1$ and $m_2$ such that $[m_1]_{\#} p_1 = [m_2]_{\#} p_1 = p_1$, defined as follows:

$$m_1 = r_1^{-1} \circ r_2$$
$$m_2 = r_2^{-1} \circ r_3,$$

$\square$

Based on lemma A.4, for a given permutation matrix $\mathbf{\Pi}$, let us define $\mathcal{H}_{\mathbf{\Pi}}$ as the set of all componentwise translation functions $\boldsymbol{h}$ such that $\boldsymbol{h}(\mathbf{\Pi}\boldsymbol{y}) \overset{d}{=} \boldsymbol{y}$. Note that $\mathcal{H}_{\mathbf{\Pi}}$ is a finite set. Specifically, $|\mathcal{H}_{\mathbf{\Pi}}| = 2^D$.

Also, define $\mathcal{R}$ as the set of all permutation matrices. Note that $|\mathcal{R}| = D!$.

> **Lemma A.2** (Zero Measure Set). The set $\mathcal{Z} = \{\boldsymbol{x} \in \mathcal{X} \mid \boldsymbol{h}(\mathbf{\Pi}\boldsymbol{x}) = \boldsymbol{x}, \boldsymbol{h} \in \mathcal{H}_{\mathbf{\Pi}}, \mathbf{\Pi} \in \mathcal{R}, \mathbf{\Pi} \neq \boldsymbol{I}\}$ has measure zero under $p_{\boldsymbol{x}}$.

*Proof.* For a given $\mathbf{\Pi} \in \mathcal{R}, \mathbf{\Pi} \neq \boldsymbol{I}$ and $\boldsymbol{h} \in \mathcal{H}_{\mathbf{\Pi}}$. Let

$$\mathcal{Z}_{\mathbf{\Pi},\boldsymbol{h}} = \{\boldsymbol{x} \in \mathcal{X} \mid \boldsymbol{h}(\mathbf{\Pi}\boldsymbol{x}) = \boldsymbol{x}\} \tag{25}$$

If we show that $\mathcal{Z}_{\mathbf{\Pi},\boldsymbol{h}}$ has measure zero under $p_{\boldsymbol{x}}$, then the set $\mathcal{Z}$ has measure zero under $p_{\boldsymbol{x}}$. This is because

$$\mathcal{Z} = \bigcup_{\mathbf{\Pi} \in \mathcal{R}, \mathbf{\Pi} \neq \boldsymbol{I}} \bigcup_{\boldsymbol{h} \in \mathcal{H}_{\mathbf{\Pi}}} \mathcal{Z}_{\mathbf{\Pi},\boldsymbol{h}}$$

and the union of finite number of measure zero sets has measure zero.

To show that $\mathcal{Z}_{\mathbf{\Pi},\boldsymbol{h}}$ has measure zero under $p_{\boldsymbol{x}}$, let us denote $\mathcal{Q} \subset [D]$ be the set of indices for which $\pi(q) \neq q, q \in \mathcal{Q}$. Let $\boldsymbol{x}_Q$ denote the vector of components of $\boldsymbol{x}$ indexed by $\mathcal{Q}$.

We can re-write $\mathcal{Z}_{\mathbf{\Pi},\boldsymbol{h}}$ as follows:

$$\mathcal{Z}_{\mathbf{\Pi},\boldsymbol{h}} = \{\boldsymbol{x} \in \mathcal{X} \mid h_q(x_{\pi(q)}) = x_q, \forall q \in \mathcal{Q}\}$$
$$= \{\boldsymbol{x} \in \mathcal{X} \mid h_q^{-1}(x_q) = x_{\pi(q)}, \forall q \in \mathcal{Q}\}$$

Take any fixed $q' \in \mathcal{Q}$. Let $\mathcal{T} = [D] \setminus \{q', \pi(q')\}$.

For any set $\mathcal{A} \subseteq R^m$, denote by $\mathcal{A}|_{x_\mathcal{B}}$ the cross-section set such that $\mathcal{A}|_{x_\mathcal{B}} = \{z \in \mathcal{A} \mid z_\mathcal{B} = x_\mathcal{B}\}$, where $x_\mathcal{B}$ is a fixed vector, and $\mathcal{B} \subseteq [m]$. Further, denote by $\mathcal{A}|^\mathcal{B}$ the projection of set $\mathcal{A}$ onto the components indexed by $\mathcal{B}$, i.e.,

$$\mathcal{A}|^\mathcal{B} = \{z_\mathcal{B} \mid z \in \mathcal{A}\}$$

.

Now, consider the following:

$$\Pr(\mathcal{Z}_{\mathbf{\Pi},\boldsymbol{h}})$$

$$= \int_{\mathcal{Z}_{\mathbf{\Pi},\boldsymbol{h}}} p_{\boldsymbol{x}}(\widetilde{\boldsymbol{x}})d\widetilde{\boldsymbol{x}}$$

$$= \int_{\mathcal{Z}_{\mathbf{\Pi},\boldsymbol{h}}} p_{\boldsymbol{x}}(\widetilde{\boldsymbol{x}}_{q'}, \widetilde{\boldsymbol{x}}_{\pi(q')}, \widetilde{\boldsymbol{x}}_{\mathcal{T}})d\widetilde{\boldsymbol{x}}_{q'}d\widetilde{\boldsymbol{x}}_{\pi(q')}d\widetilde{\boldsymbol{x}}_{\mathcal{T}}$$

$$= \int_{R^D} I([\widetilde{\boldsymbol{x}}_{q'}, \widetilde{\boldsymbol{x}}_{\pi(q')}, \widetilde{\boldsymbol{x}}_{\mathcal{T}}] \in \mathcal{Z}_{\mathbf{\Pi},\boldsymbol{h}})\, p_{\boldsymbol{x}}(\widetilde{\boldsymbol{x}}_{q'}, \widetilde{\boldsymbol{x}}_{\pi(q')}, \widetilde{\boldsymbol{x}}_{\mathcal{T}})d\widetilde{\boldsymbol{x}}_{q'}d\widetilde{\boldsymbol{x}}_{\pi(q')}d\widetilde{\boldsymbol{x}}_{\mathcal{T}}$$

$$= \int_{R^{D-1}} I([\widetilde{\boldsymbol{x}}_{q'}, \widetilde{\boldsymbol{x}}_{\mathcal{T}}] \in \mathcal{Z}_{\mathbf{\Pi},\boldsymbol{h}}|^{\mathcal{T}\cup\{q'\}})\, p_{\boldsymbol{x}_{[D]\setminus\{q'\}}}(\widetilde{\boldsymbol{x}}_{\mathcal{T}}, \widetilde{\boldsymbol{x}}_{q'}) \times$$

$$\left\{ \int_R I(\widetilde{\boldsymbol{x}}_{\pi(q')} \in \mathcal{Z}_{\mathbf{\Pi},\boldsymbol{h}}|_{\widetilde{\boldsymbol{x}}_{q'}, \widetilde{\boldsymbol{x}}_{\mathcal{T}}})\, p_{x_{\pi(q')}}(\widetilde{x}_{\pi(q')} \mid \widetilde{x}_{q'}, \widetilde{\boldsymbol{x}}_{\mathcal{T}})d\widetilde{x}_{\pi(q')} \right\} d\widetilde{\boldsymbol{x}}_{q'}d\widetilde{\boldsymbol{x}}_{\mathcal{T}}$$

$$= \int_{R^{D-1}} I([\widetilde{\boldsymbol{x}}_{q'}, \widetilde{\boldsymbol{x}}_{\mathcal{T}}] \in \mathcal{Z}_{\mathbf{\Pi},\boldsymbol{h}}|^{\mathcal{T}\cup\{q'\}})\, p_{\boldsymbol{x}_{[D]\setminus\{q'\}}}(\widetilde{\boldsymbol{x}}_{\mathcal{T}}, \widetilde{\boldsymbol{x}}_{q'}) \times$$

$$\underbrace{\left\{ \int_R I(\widetilde{x}_{\pi(q')} = h_{q'}^{-1}(\widetilde{x}_{q'}))\, p_{x_{\pi(q')}}(\widetilde{x}_{\pi(q')} \mid \widetilde{x}_{q'}, \widetilde{\boldsymbol{x}}_{\mathcal{T}})d\widetilde{x}_{\pi(q')} \right\}}_{=0} d\widetilde{\boldsymbol{x}}_{q'}d\widetilde{\boldsymbol{x}}_{\mathcal{T}}$$

$$= 0.$$

The reason the inner integral is zero is as follows. $\widetilde{x}_{q'}$ is fixed for the inner integral and $p_{x_{\pi(q')}}$ is finite over its support. Therefore $I(\widetilde{x}_{\pi(q')} = h_{q'}^{-1}(\widetilde{x}_{q'})) = 0$ almost everywhere except on the singleton set $\{h_{q'}^{-1}(\widetilde{x}_{q'})\}$, which has measure zero.

Finally, since $\mathcal{Z}$ is a finite union of measure zero sets, it has measure zero. $\square$

## B. Proof of Proposition 4.1

> **Proposition (Gaussian sparse probes yield a tight proxy for $\|\boldsymbol{J}\|_0$).** Let $\boldsymbol{J} \in \mathbb{R}^{D \times D}$. For each row $d \in \{1, \dots, D\}$, define the support $\mathcal{T}_d := \{j \in \{1, \dots, D\} : \boldsymbol{J}_{dj} \neq 0\}$ and its size $T_d := |\mathcal{T}_d|$. Fix an integer $1 \leq S \leq D$, and draw a random mask $\boldsymbol{r} \sim \mathrm{Unif}(\mathcal{R}_{D,S})$, where $\mathcal{R}_{D,S} = \{\boldsymbol{r} \in \{0,1\}^D \mid \|\boldsymbol{r}\|_0 = S\}$. Independently draw $\boldsymbol{\epsilon} \sim \mathcal{N}(0, I_D)$, and define the probe $\boldsymbol{z} := \boldsymbol{r} \odot \boldsymbol{\epsilon}$. Define
>
> $$\boldsymbol{q}_{\mathrm{G}}(\boldsymbol{J}) := \frac{D}{S}\, \mathbb{E}_{\boldsymbol{r}, \boldsymbol{\epsilon}} \|\boldsymbol{J}\boldsymbol{z}\|_0.$$
>
> Then $\boldsymbol{q}_{\mathrm{G}}(\boldsymbol{J})$ satisfies the sandwich bound
>
> $$\|\boldsymbol{J}\|_0 \;\geq\; \boldsymbol{q}_{\mathrm{G}}(\boldsymbol{J}) \;\geq\; \left(1 - \frac{(S-1)(T-1)}{2(D-1)}\right) \|\boldsymbol{J}\|_0, \qquad T := \max_d T_d.$$

*Proof.* For each row $d \in \{1, \dots, D\}$, define the indicator variable

$$X_d := \mathbf{1}\big[(\boldsymbol{J}\boldsymbol{z})_d \neq 0\big].$$

Then

$$\|\boldsymbol{J}\boldsymbol{z}\|_0 = \sum_{d=1}^D X_d, \qquad \mathbb{E}_{\boldsymbol{r},\boldsymbol{\epsilon}}\|\boldsymbol{J}\boldsymbol{z}\|_0 = \sum_{d=1}^D \mathbb{E}_{\boldsymbol{r},\boldsymbol{\epsilon}}X_d = \sum_{d=1}^D \mathbb{P}\big((\boldsymbol{J}\boldsymbol{z})_d \neq 0\big).$$

Fix a row $d$ and abbreviate $\mathcal{T} := \mathcal{T}_d$ and $T := |\mathcal{T}| = T_d$. Let $\mathcal{Z}$ be the (random) support of $\boldsymbol{r}$ (so $|\mathcal{Z}| = S$), and define

$$B := |\mathcal{T} \cap \mathcal{Z}|.$$

Since $z_j = r_j \epsilon_j$ with $r_j \in \{0, 1\}$, we have

$$(\boldsymbol{Jz})_d = \sum_{j=1}^{D} \boldsymbol{J}_{dj} z_j = \sum_{j \in \mathcal{T} \cap \mathcal{Z}} \boldsymbol{J}_{dj}\, \epsilon_j.$$

Conditioned on $\mathcal{Z}$, $(\boldsymbol{Jz})_d$ is Gaussian:

$$(\boldsymbol{Jz})_d \,\big|\, \mathcal{Z} \sim \mathcal{N}\big(0,\ \sigma^2(\mathcal{Z})\big), \qquad \sigma^2(\mathcal{Z}) := \sum_{j \in \mathcal{T} \cap \mathcal{Z}} \boldsymbol{J}_{dj}^2.$$

If $B = 0$, then $\sigma^2(\mathcal{Z}) = 0$ and $(\boldsymbol{Jz})_d = 0$ a.s. If $B \geq 1$, then $\sigma^2(\mathcal{Z}) > 0$ (since $\boldsymbol{J}_{dj} \neq 0$ on $\mathcal{T}$), hence

$$\mathbb{P}\big((\boldsymbol{Jz})_d = 0 \,\big|\, \mathcal{Z}\big) = 0.$$

Therefore,

$$\mathbf{1}\big[(\boldsymbol{Jz})_d \neq 0\big] = \mathbf{1}[B \geq 1] \quad \text{a.s.} \quad \Rightarrow \quad \mathbb{P}\big((\boldsymbol{Jz})_d \neq 0\big) = \mathbb{P}(B \geq 1).$$

To bound $\mathbb{P}(B \geq 1)$, for each $j \in \mathcal{T}$ define the event $A_j := \{j \in \mathcal{Z}\}$. Then $\{B \geq 1\} = \bigcup_{j \in \mathcal{T}} A_j$. Because $\mathcal{Z}$ is a uniform subset of size $S$,

$$\mathbb{P}(A_j) = \frac{S}{D} \quad \text{for all } j, \qquad \mathbb{P}(A_j \cap A_\ell) = \mathbb{P}(\{j, \ell\} \subseteq \mathcal{Z}) = \frac{S(S-1)}{D(D-1)} \quad \text{for } j \neq \ell.$$

By the union bound,

$$\mathbb{P}(B \geq 1) = \mathbb{P}\Big( \bigcup_{j \in \mathcal{T}} A_j \Big) \leq \sum_{j \in \mathcal{T}} \mathbb{P}(A_j) = T \cdot \frac{S}{D}.$$

By inclusion–exclusion up to second order,

$$\mathbb{P}\Big( \bigcup_{j \in \mathcal{T}} A_j \Big) \geq \sum_{j \in \mathcal{T}} \mathbb{P}(A_j) - \sum_{\substack{j, \ell \in \mathcal{T} \\ j < \ell}} \mathbb{P}(A_j \cap A_\ell) = T \cdot \frac{S}{D} - \binom{T}{2} \frac{S(S-1)}{D(D-1)}.$$

Therefore, for each row $d$,

$$T_d \frac{S}{D} - \binom{T_d}{2} \frac{S(S-1)}{D(D-1)} \ \leq\ \mathbb{P}\big((\boldsymbol{Jz})_d \neq 0\big) \ \leq\ T_d \frac{S}{D}. \tag{26}$$

Summing (26) over $d = 1, \ldots, D$ yields

$$\frac{S}{D} \sum_{d=1}^{D} T_d - \frac{S(S-1)}{D(D-1)} \sum_{d=1}^{D} \binom{T_d}{2} \ \leq\ \mathbb{E}_{\boldsymbol{r}, \boldsymbol{\epsilon}} \|\boldsymbol{Jz}\|_0 \ \leq\ \frac{S}{D} \sum_{d=1}^{D} T_d.$$

Noting that $\sum_{d=1}^{D} T_d = \|\boldsymbol{J}\|_0$, we obtain

$$\|\boldsymbol{J}\|_0 - \frac{(S-1)}{(D-1)} \sum_{d=1}^{D} \binom{T_d}{2} \ \leq\ \frac{D}{S} \mathbb{E}_{\boldsymbol{r}, \boldsymbol{\epsilon}} \|\boldsymbol{Jz}\|_0 \ \leq\ \|\boldsymbol{J}\|_0.$$

Finally, $T_d \leq T := \max_d T_d$ implies

$$\binom{T_d}{2} = \frac{T_d(T_d - 1)}{2} \leq \frac{T-1}{2} T_d,$$

so

$$\sum_d \binom{T_d}{2} \leq \frac{T-1}{2} \sum_d T_d = \frac{T-1}{2} \|\boldsymbol{J}\|_0,$$

which gives the claimed sandwich bound. $\qquad\square$

## C. Alternative Identifiability Conditions Beyond Jacobian Sparsity

Assumption 3.1 (Jacobian sparsity) plays the role of an identifiability condition for the underlying nonlinear unmixing step (Step 1 of the proof of Theorem 3.4). As with most identifiability theories, this assumption characterizes a meaningful regime in which the proposed criterion is provably valid; it is not intended to cover every possible data-generating process. We argue below that (i) the assumption is well-motivated for many practical DT settings, and (ii) the proposed identifiability framework is modular, in the sense that it can accommodate other unmixing identifiability conditions when sparsity does not hold.

Assumption 3.1 make sense when the cross-domain feature interactions are sparse or local in nature. Concrete cases include: (a) geometric transformations such as rotations, translations, and warping (each output coordinate depends on a small neighborhood of input coordinates); (b) inverse problems such as super-resolution, denoising, and inpainting, where each output pixel/feature is influenced by a limited spatial neighborhood; (c) sparse data representations (e.g., gene-expression or single-cell modalities) where each output feature is driven by a small subset of latent factors; and (d) language transduction, where the target token typically depends on a limited window of source tokens. In all these cases, the Jacobian of the ground-truth transfer map can be reasonably modeled as having sparse row supports.

**Alternative unmixing conditions.**    When Jacobian sparsity does not hold, identifiability can still be obtained through alternative structural conditions on the mixing function. The nonlinear-unmixing literature has established several such conditions, including *diverse influence* (Nguyen & Fu, 2025), *independent mechanisms/orthogonal Jacobians* (Gresele et al., 2021), and other function-class-based conditions (Buchholz et al., 2022). Each of these can replace Step 1 of our proof: the criterion in Eq. (7) would simply use the corresponding regularizer (e.g., a Jacobian-orthogonality penalty in place of $\|J_g\|_0$), with Steps 2 and 3 (distribution matching and the anchor argument) unchanged. A systematic study of these alternative conditions in the DT context is beyond the scope of this work, but our analytical framework is well-suited to incorporate them.

## D. Detailed Derivation of the Sparsity Regularizer

The sparsity regularizer used in our high-dimensional implementation,

$$\mathcal{L}_{\mathrm{sp}}(g_\theta) = \mathbb{E}_{x,z}\left\|\frac{g_\theta(x + \delta z) - g_\theta(x)}{\delta}\right\|_1, \tag{27}$$

is a chain of three approximations of the exact Jacobian-$\ell_0$ objective $\mathbb{E}_x\|J_{g_\theta}(x)\|_0$ that appears in Eq. (7a). We make each step explicit here for clarity.

**Step 1: From $\|J\|_0$ to a Jacobian-vector-product proxy.**    Computing $\|J\|_0$ in high dimensions requires forming the full $D \times D$ Jacobian, which is prohibitive. Proposition 4.1 shows that for a sparse Gaussian probe $z = r \odot \epsilon$ with mask sparsity $\|r\|_0 = S$, the quantity $q(J) = (D/S)\mathbb{E}_z\|Jz\|_0$ sandwiches $\|J\|_0$:

$$\|J\|_0 \ \geq \ q(J) \ \geq \ \left(1 - \frac{(S-1)(T-1)}{2(D-1)}\right)\|J\|_0.$$

Therefore, $q(J)$ is a tight proxy whenever $(S-1)(T-1) \ll D$. Roughly, instead of evaluating the support of every column of $J$, it is enough to evaluate the support of the action of $J$ on a few sparse random probes. Replacing $\|J\|_0$ by $q(J)$ in Eq. (7a) produces an objective of the form $\mathbb{E}_{x,z}\|J_{g_\theta}(x)z\|_0$ (up to a constant).

**Step 2: From $\ell_0$ to $\ell_1$.**    The $\ell_0$ pseudo-norm is non-differentiable and not amenable to gradient-based optimization. Following standard practice in sparse signal processing, we relax it to its convex surrogate $\ell_1$, yielding the objective $\mathbb{E}_{x,z}\|J_{g_\theta}(x)z\|_1$.

**Step 3: From Jacobian-vector products to finite differences.**    For a smooth $g_\theta$ and small $\delta > 0$, the directional derivative is well-approximated by a forward finite difference:

$$J_{g_\theta}(x)z \ = \ \lim_{\delta \to 0}\frac{g_\theta(x + \delta z) - g_\theta(x)}{\delta} \ \approx \ \frac{g_\theta(x + \delta z) - g_\theta(x)}{\delta}.$$

Substituting this into the previous step gives Eq. (27).

Consequently, each evaluation of Eq. (27) requires only *two forward passes* of $g_\theta$ per sample, regardless of $D$. With $S > 1$, a few Monte-Carlo samples of $z$ suffice to obtain a low-variance estimate (see Section E.5), in contrast to the standard column-wise finite difference which requires $\mathcal{O}(D)$ probes. Hence Eq. (27) is a faithful and scalable surrogate for $\mathbb{E}_x \|J_{g_\theta}(x)\|_0$.

# E. Experiments

## E.1. Setting

### E.1.1. 2D Synthetic Experiment

Hyperparameter settings for 2D synthetic experiment is detailed in Table 2.

*Table 2.* Hyperparameter settings for 2D synthetic experiments.

| Parameter | Value |
|---|---|
| *Optimization* | |
| Learning rate | $10^{-3}$ |
| Batch size | 1024 |
| Training iterations | 7,000 |
| *Data* | |
| Number of training samples | 27,000 |
| Number of test samples | 3,000 |
| *Loss Weights* | |
| $\lambda_{\text{inv}}$ | 1.0 |
| $\lambda_{\text{sp}}$ | 0.1 |
| $\lambda_{\text{anch}}$ | 1.0 |
| $|\mathcal{E}|$ | 1 |

**Neural Network Architecture.** The translation network $g_\theta$ and reconstruction network $f_\alpha$ are implemented as 2-layer multi-layer perceptrons (MLPs), each with 32 hidden units per layer and LeakyReLU activations (negative slope 0.2). The discriminator is also a 2-layer MLP but with 64 hidden units per layer and LeakyReLU activations. All networks are optimized using Adam with a learning rate of $10^{-3}$.

### E.1.2. MNIST to rotated MNIST Experiment

Hyperparameter settings for MNIST to rotated MNIST experiment is detailed in Table 3.

**Neural Network Architecture.** We use fully connected networks for both the translation network $g_\theta$ and discriminator $d_\psi$. The translation network is implemented as a two-layer MLP that first flattens the $32 \times 32$ input image into a vector, applies a linear transformation to a 1024-dimensional hidden layer with ReLU activation, followed by another linear layer with Tanh activation, and finally reshapes the output to a $32 \times 32$ image. The discriminator follows a similar architecture: flattening the input, applying a linear layer to 1024 hidden units with LeakyReLU activation (negative slope 0.2), and a final linear layer producing a single scalar output for real/fake classification. We use the non-saturating GAN loss.

### E.1.3. Edges to rotated shoes Experiment

Hyperparameter settings for Edges to rotated shoes experiment is detailed in Table 4.

**Neural Network Architecture.** Input images are resized to $128 \times 128$ and are encoded by the frozen VAE encoder of Stable Diffusion (Rombach et al., 2022) into $16 \times 16 \times 4$ for training efficiency. The translation network uses a StarGAN-style encoder-decoder architecture (Choi et al., 2020) built from residual blocks, where the encoder consists of an initial

*Table 3.* Hyperparameter settings for MNIST experiments.

| Parameter | Value |
|---|---|
| *Optimization* | |
| Learning rate | $10^{-4}$ |
| Optimizer | Adam ($\beta_1 = 0.0$, $\beta_2 = 0.99$) |
| Weight decay | $10^{-5}$ |
| Batch size | 16 |
| Training iterations | 50,000 |
| *Loss Weights* | |
| $\lambda_{\mathrm{inv}}$ | 1.0 |
| $\lambda_{\mathrm{sp}}$ | 0.1 |
| $\lambda_{\mathrm{anch}}$ | 1.0 |
| $|\mathcal{E}|$ | 1 |
| *Jacobian Regularization* | |
| Method | Masked Finite-difference (Eq. (20)) |
| Number of $z$ samples for approximating $\mathbb{E}_{z}\|J_{g_{\theta}}(x)z\|_1$ | 8 |
| $S$ | 102 |
| $\delta$ | 0.01 |

convolution and 2 downsampling residual blocks, and the decoder contains two upsampling residual blocks with AdaIN (Adaptive Instance Normalization) layers. As discussed in Section 5.2, we replace standard instance normalization with channel-only normalization to keep the Jacobian sparsity regularization compatible with the network. The discriminator is a convolutional network that mirrors the encoder with residual blocks for downsampling and outputs a single scalar for real/fake classification. To stabilize GAN training, we apply R1 regularization (Mescheder et al., 2018) to the discriminator network with a weight of 3. During evaluation, the translation is performed in latent space, and the result is decoded to image space using the frozen VAE decoder. The reconstruction network $f_{\alpha}$ shares the same architecture as the translation network.

### E.2. Ablation Study

We conduct an ablation study on the MNIST to rotated MNIST task to understand the contribution of each component in our proposed method. We compare four configurations by enabling/disabling Jacobian sparsity regularization ($\lambda_{\mathrm{sp}}$) and anchor matching regularization ($\lambda_{\mathrm{anch}}$).

- Proposed: $\lambda_{\mathrm{sp}} = 0.01$, $\lambda_{\mathrm{anch}} = 1$,
- Case I: $\lambda_{\mathrm{sp}} = 0$, $\lambda_{\mathrm{anch}} = 0$,
- Case II: $\lambda_{\mathrm{sp}} = 0.01$, $\lambda_{\mathrm{anch}} = 0$,
- Case III: $\lambda_{\mathrm{sp}} = 0$, $\lambda_{\mathrm{anch}} = 1$.

**Results:** Table 5 shows the TE attained in various cases. It shows that only using anchor matching and sparsity regularization jointly with distribution matching results in acceptable TE. Fig. 8 shows the translation attained in various cases. It is clear that both regularizations are necessary to obtain content-aligned DT.

### E.3. Effect of the Number of Anchors

Table 6 reports LPIPS on Edges to rotated Shoes as we vary the number of paired anchors $|\mathcal{E}| \in \{1, 2, 5, 10\}$. Performance improves monotonically with more anchors, with the largest gain when going from 5 to 10. This is consistent with the discussion in Section 5.2.2: small numbers of anchors can be insufficient under model mismatch and imperfect optimization in this high-dimensional regime, but a modest set of anchors recovers strong content alignment.

*Table 4.* Hyperparameter settings for Edges2Shoes experiments.

| Parameter | Value |
|---|---|
| *Optimization* | |
| Learning rate | $10^{-4}$ |
| Optimizer | Adam ($\beta_1 = 0.0$, $\beta_2 = 0.99$) |
| Weight decay | $10^{-5}$ |
| Batch size | 16 |
| Training iterations | 100,000 |
| *Loss Weights* | |
| $\lambda_{\text{inv}}$ | 10.0 |
| $\lambda_{\text{anch}}$ | 10.0 |
| $\lambda_{\text{sp}}$ | 0.001 |
| $|\mathcal{E}|$ | 10 |
| *Jacobian Regularization* | |
| Method | Masked Finite-difference (Eq. (20)) |
| Number of $z$ samples for approximating $\mathbb{E}_z \| J_{g_\theta}(x) z \|_1$ | 16 |
| $S$ | 20 |
| $\delta$ | 0.01 |

*Table 5.* Ablation study on MNIST $\rightarrow$ rotated MNIST. Translation error (TE) measures content alignment; lower is better.

| Configuration | TE($\downarrow$) |
|---|---|
| Distr. Matching (Case I) | $0.50 \pm 0.19$ |
| + Sparsity reg. (Case II) | $0.63 \pm 0.19$ |
| + Anchor (Case III) | $0.65 \pm 0.12$ |
| + Anchor + Sparsity reg. (Proposed) | $\mathbf{0.20 \pm 0.06}$ |

### E.4. Sensitivity to Anchor Choice

A natural concern raised during reviewing is that, with a single anchor, the learned mapping may be overly sensitive to the particular anchor pair drawn. To test this, we run the proposed method on the MNIST to rotated MNIST task across 5 independent trials, each using a different randomly sampled anchor pair $(x^{(\ell)}, y^{(\ell)})$. To control runtime, each trial is trained for 20,000 iterations rather than the 50,000 used elsewhere. We report the average translation error (TE) on the test set across trials.

The mean test TE across the 5 trials is $0.367$, with standard deviation $0.094$. The variation across anchors is therefore limited, indicating that the proposed criterion is not strongly sensitive to the specific choice of anchor sample within reasonable variability.

### E.5. Jacobian Sparsity Regularization

**Variance Reduction Effect** Since the proposed masked finite-difference based regularizer in Eq. (20) takes into consideration $S$ columns of Jacobian at once, it is expected to have lower variance than column-wise finite-difference based regularizer, i.e., when $S = 1$. We can also observe this fact in Fig. 9. Here we generate 20 Jacobian matrices $J$ of dimensions $D \times D$ with $D = 1000$ and $T = 10$. The non-zero entries of $J$ are sampled from $\mathcal{N}(0, 1)$. We plot the bias and variance of $\frac{D}{S} \| J z \|_0$ as a function of $S$. We use 500 Monte Carlo samples of $z$ in order to estimate the variance. The plot shows the average variance and relative bias of the estimator for all Jacobians. As expected the variance seems to decrease exponentially with increasing $S$, whereas the bias increases linearly with $S$. Hence it is desirable to use a moderate $S$ for the masked finite-difference based regularizer.

**Effect of $S$ on MNIST to rotated MNIST** Fig. 10 shows the effect of using $S = 1$ and $S = 100$ for MNIST to rotated MNIST dataset. One can see that when $S = 1$, which corresponds to column-wise sampling of the Jacobian, results in

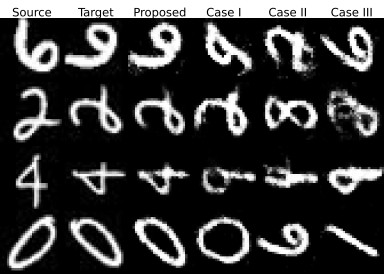

*Figure 8.* Ablation study on MNIST → rotated MNIST. Case I: only distribution matching, Case II: sparsity regularization + distribution matching, Case III: anchor alignment + distribution matching, Case IV: anchor alignment + sparsity regularization + distribution matching.

*Table 6.* LPIPS on Edges to rotated Shoes as a function of the number of paired anchors $|\mathcal{E}|$.

| # Aligned samples $|\mathcal{E}|$ | LPIPS($\downarrow$) |
| :---: | :---: |
| 1 | $0.400 \pm 0.09$ |
| 2 | $0.390 \pm 0.10$ |
| 5 | $0.379 \pm 0.09$ |
| 10 | $\mathbf{0.320 \pm 0.07}$ |

misaligned translation, whereas using $S = 100$ is significantly better at alignment.

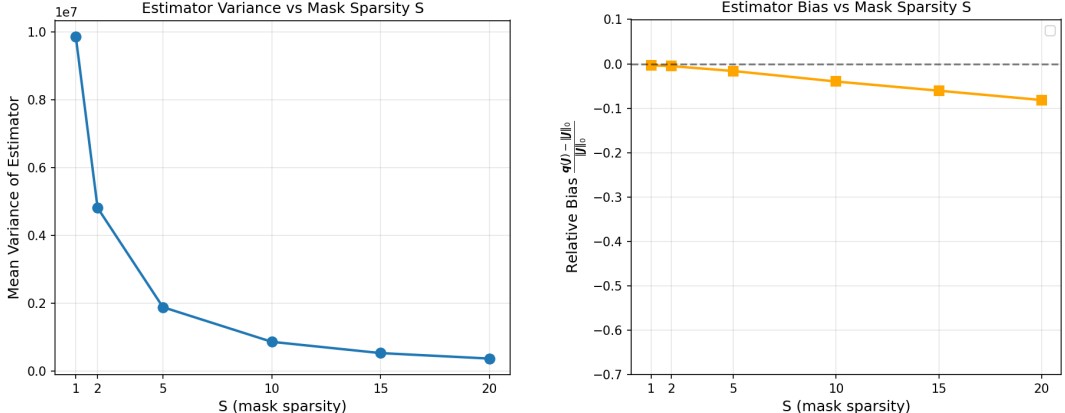

*Figure 9.* Variance (a) and bias (b) of $\frac{D}{S}\|\boldsymbol{Jz}\|_0$ as a function of $S$. Note that $S = 1$ corresponds to the column-wise finite-difference based regularizer in (17).

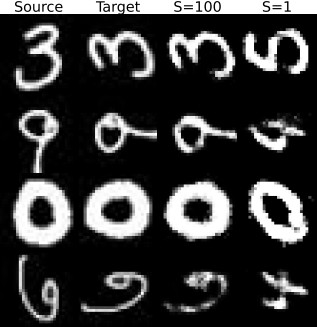

*Figure 10.* MNIST to Rotated MNIST using finite-masked difference with $S = 1$ and $S = 100$.

