# OpenReview forum: "Domain Transfer Becomes Identifiable via a Single Alignment"
_ICML.cc/2026/Conference — ICML 2026 regular_

### Official Review · Reviewer_GxmU · 2026-02-15

**Soundness:** 3
**Presentation:** 3
**Significance:** 2
**Originality:** 3
**Overall Recommendation:** 4
**Confidence:** 5

**Summary:**

This paper studies the identifiability problem in unsupervised domain transfer (DT). Standard distribution matching approaches are ill-posed because measure-preserving automorphisms can produce content-misaligned mappings while preserving marginal distributions. The authors propose a new route to identifiability that requires only a single paired anchor sample under a structural sparsity assumption on the Jacobian support of the transfer function.

**Compliance With Llm Reviewing Policy:**

Affirmed.

**Ethical Review Concerns:**

Not applicable.

**Key Questions For Authors:**

See weaknesses.

**Limitations:**

1. Expand the limitations beyond “anchors may be unavailable” and “sparsity may not hold,” and include failure modes when assumptions are violated (dense Jacobians, imperfect invertibility, mode collapse, or content misalignment despite matching).

2. Add robustness analysis for noisy/mismatched anchor pairs, including sensitivity to anchor choice and the effect of using multiple anchors versus one.

**Strengths And Weaknesses:**

# Strengths
1. The non-identifiability of unsupervised domain transfer is well known yet poorly understood. This work tackles a core theoretical limitation with clear practical implications.

2. The idea of "single aligned pair can eliminate ambiguity under structural sparsity" is interesting and potentially impactful.

## Weaknesses
1. The identifiability guarantee relies on a Jacobian sparsity condition (Assumption 3.1) that may not hold for many real-world domain transfer tasks. The practical scope of tasks that satisfy this assumption is unclear.

2. The experiments focus on relatively simple transformations (e.g., rotations). It is unclear whether the method scales to complex semantic changes such as style transfer, medical imaging, or cross-modal translation.

3. The experiments are mostly conducted on MNIST and Edges-to-Shoes, which are relatively simplified toy datasets. Evaluation on more realistic datasets is needed.

4. The paper does not deeply analyze scenarios where sparsity assumptions are violated or anchors are noisy.  In practice, a single point cannot constrain complex nonlinear functions. If the anchor is atypical or noisy, it may bias the mapping.

5. Distribution matching constraint does not ensure semantic alignment, as noted. However, the paper could more clearly explain this failure mode.

---

> ### Author Rebuttal · Authors · 2026-03-31
>
> &nbsp;
>
> **Anonymous link** : [(Click Here)](https://drive.google.com/file/d/1aA_lPh4Qe_ypDaS3raxeceooLjKcIUt0/view?usp=sharing)
>
> &nbsp;
>
> **[Jacobian Sparsity holds for all datasets ?]**
>
> Indeed, Assumption 3.1 cannot hold for all real-world DT datasets in the DT field. We also acknowledge this in the limitations. Nonetheless, the analytical framework can also work with a broader range of assumptions on $g^\\star$. Let us explain.
>
> As in most theoretical works, the role of assumption 3  is to characterize a meaningful regime where the method is provably valid, not to cover every possible data-generating process. In our paper, Assumption 3.1 is motivated by settings where the cross-domain feature interaction is sparse or local such as geometric transformations, sparse data representations, inverse problems (e.g., super-resolution, inpainting, denoising), and language processing (output token depending upon limited number of input tokens)
>
> More importantly, when Jacobian sparsity does not hold, identifiability may still be obtained via alternative assumptions. In our theory, Assumption 3.1 is used to guarantee identifiable nonlinear unmixing. However, prior work has established alternative unmixing results under different conditions, such as diverse influence [R1] or orthogonal Jacobians [R2, R3]. These results may apply to DT settings not covered by Assumption 3.1, and in principle could be incorporated into our framework by replacing the sparsity regularization with the corresponding alternative regularization. A systematic study of such alternative unmixing conditions is beyond the scope of this paper, but we will add an appendix discussion in the revised version to clarify these alternatives to the Jacobian sparsity assumption.
>
> [R1] Nguyen et. al. Diverse Influence Component Analysis: A Geometric Approach to Nonlinear Mixture Identifiability, NeurIPS 2025
>
> [R2] Gresele et al., Independent mechanism analysis, a new concept. NeurIPS 2021
>
> [R3] Buchholz et al., "Function classes for identifiable nonlinear independent component analysis." NeurIPS 2022.
>
> &nbsp;
>
> **[More complex Dataset]**
>
> To address the reviewer's concern, we include additional experiment on single cell sequence alignment, an important problem in biomedical research. Due to space constraint, please refer to the response to Reviewer pvLG for the details of the experiments with results.
>
> &nbsp;
>
> **[Sparsity violation and Noisy Anchor Analysis]**
>
> As we discussed, the impact of sparsity violation could be compensated by using other unmixing techniques, e.g., those in [R1, R2]. Noisy anchor samples (such as approximate pair, instead of exact pair)’s impact is indeed unclear under our current analysis. In practice, this limitation could be remedied readily by obtaining more than one aligned pairs. However, a theoretical analysis is indeed of interest. Nonetheless, approximate identifiability, robustness analysis, and finite sample analysis generally require completely different computational tools and proof techniques. As such, these analyses usually require a dedicated research effort in the identifiability literature (e.g., nonlinear unmixing [R1], and final sample analysis [R2]). We will add this as a limitation and an important future work.
>
> [R1] Hyvarinen et al., Nonlinear ICA using auxiliary variables and generalized contrastive learning. AISTATS 2019.
>
> [R2] Lyu et al., On Finite-Sample Identifiability of Contrastive Learning-Based Nonlinear Independent Component Analysis. ICML 2022.
>
> &nbsp;
>
> **[Sensitivity to anchor choice]**
>
> To observe sensitivity with respect to the choice of anchor, we run 5 trials of the proposed method for MNIST to rotated MNIST dataset with different random choice of anchor in each trial. We train the model for 20k iterations only to control the run-time (full training in the paper is 50k iterations) and report the mean and std of the average TE for the test set. The mean is 0.367, and the std is 0.094, which suggests a limited variation due to the choice of anchor.
>
> &nbsp;
>
> **[Multiple anchors vs one]**
>
> To alleviate the reviewer’s concern, we conduct an ablation study with number of anchors for the Edges to rotated Shoes dataset, where multiple anchors was needed. Table 1 in the anonymized link shows the result. From the table, <= 5 anchors does not appear to be insufficient for a reasonable alignment performance.
>
> &nbsp;
>
> **[Further explanation of Failure Mode]**
>
> We had explained about the failure mode in Section 2.1 under MPAs and non-identifiability with illustrations in Fig. 1 and Fig. 2 of the manuscript. The failure mode is also well documented in the literature [R3, R4]. Nonetheless, we will add an appendix section with more details and references in the revised version.
>
> [R3] Moriakov et al., Kernel of CycleGAN as a principle homogeneous space. ICLR 2020
>
> [R4] Shrestha et al.,  "Towards identifiable unsupervised domain translation: A diversified distribution matching approach." ICLR 2024.

---

> > ### Author Rebuttal · Reviewer_GxmU · 2026-04-04
> >
> > Thanks. I would maintain my score according to the evaluation of the feedbacks.

---

### Official Review · Reviewer_iuhq · 2026-03-11

**Soundness:** 3
**Presentation:** 3
**Significance:** 3
**Originality:** 3
**Overall Recommendation:** 4
**Confidence:** 3

**Summary:**

This paper proposes a novel method to eliminate MPAs in the data transfer (DT) problem based on the Jacobian sparsity assumption. This method requires only less supervision information. Experiments were conducted on the handwritten character problem.

**Compliance With Llm Reviewing Policy:**

Affirmed.

**Final Justification:**

The authors have almost resolved my concerns by clarifying the scope of the Jacobian sparsity assumption, discussing alternative identifiability conditions, adding a more realistic dataset experiment, and explaining the gap between the ideal theory and practical multi-anchor setting. Thus, I would like to raise my score.

**Key Questions For Authors:**

1. Does the Jacobian matrix sparsity assumption (Assumption 3.1 in the paper) hold true for all real-world datasets in the DT field?

2. In the contribution section 1.2, the authors mention requiring only a few as a single paired “anchor” sample. However, in more complex task experiments (e.g., Edges to rotated shoes in 5.2.2), they state that 10 anchor samples were used for the proposed method. This suggests that in the presence of noise or complex manifolds, the robustness of a single anchor may be far lower than theoretically expected. The authors should explain the underlying reason for the increased supervision.

3. Issues with formulas and symbol definitions: For example, in the last sentence of the Notation in Section 1.2, the letter 'i' is used as both a set index and a set element variable, which can easily lead to misunderstandings.

4. There are spelling errors in the paper and appendices. For example, "since" is written as "sice." The authors need to check these errors.

**Limitations:**

yes

**Strengths And Weaknesses:**

Strengths:
The theoretical assumptions and reasoning process are logically rigorous, and have a certain degree of innovation and experimental value.

Weaknesses:
This work presents certain issues regarding experimental validation and methodological limitations. The authors need to supplement their work with broader real-world DT (Data Transformation) experiments; current experiments are limited to handwritten digit recognition applications. However, DT problems, especially the derivative problem MPAs (measure-preserving automorphisms) mentioned in the paper, are not limited to this field. They also exist in other areas such as medical image transfer. Currently, the method only proves effective for handwritten digit problems (2D synthetic data is more like a toy model), and its applicability to other research areas remains unclear. Furthermore, the paper primarily utilizes the sparsity of the Jacobian matrix to impose a constraint on the mapping space in such real-world problems, assuming that real-world transformations are often "local" and "simple," and should not involve complex global disorder. However, if complex real-world problems do not satisfy this assumption, this method may fail. This further suggests that the authors need to conduct a broader investigation into whether real-world DT problems typically satisfy this assumption and to perform more extensive experimental validation.

---

> ### Author Rebuttal · Authors · 2026-03-31
>
> &nbsp;
>
> **Anonymous link** : [(Click Here)](https://drive.google.com/file/d/1aA_lPh4Qe_ypDaS3raxeceooLjKcIUt0/view?usp=sharing)
>
> &nbsp;
>
> **[Jacobian Sparsity holds for all datasets ?]**
>
> Indeed, Assumption 3.1 cannot hold for all real-world DT datasets in the DT field. We also acknowledge this in the limitations. Nonetheless, the analytical framework can also work with a broader range of assumptions on $g^\\star$. Let us explain.
>
> As in most theoretical works, the role of assumption 3  is to characterize a meaningful regime where the method is provably valid, not to cover every possible data-generating process. In our paper, Assumption 3.1 is motivated by settings where the cross-domain feature interaction is sparse or local such as geometric transformations, sparse data representations, inverse problems (e.g., super-resolution, inpainting, denoising), and language processing (output token depending upon limited number of input tokens)
>
> More importantly, when Jacobian sparsity does not hold, identifiability may still be obtained via alternative assumptions. In our theory, Assumption 3.1 is used to guarantee identifiable nonlinear unmixing. However, prior work has established alternative unmixing results under different conditions, such as diverse influence [R1] or orthogonal Jacobians [R2, R3]. These results may apply to DT settings not covered by Assumption 3.1, and in principle could be incorporated into our framework by replacing the sparsity regularization with the corresponding alternative regularization. A systematic study of such alternative unmixing conditions is beyond the scope of this paper, but we will add an appendix discussion in the revised version to clarify these alternatives to the Jacobian sparsity assumption.
>
> [R1] Nguyen et. al. Diverse Influence Component Analysis: A Geometric Approach to Nonlinear Mixture Identifiability, NeurIPS 2025
>
> [R2] Gresele et al., Independent mechanism analysis, a new concept. NeurIPS 2021
>
> [R3] Buchholz et al., "Function classes for identifiable nonlinear independent component analysis." NeurIPS 2022.
>
> &nbsp;
>
> **[More complex Dataset]**
>
> To further show the usefulness of the proposed framework in real-world applications, we run an experiment on a single-cell sequence alignment problem. The task aims to align single cell sequence measurements captured by different modalities (such as RNA-seq and ATAC-seq). The problem stems from the fact that single cell measurement processes are destructive, hence obtaining paired measurements is difficult. This is a core task in biological data analysis and AI4S [R1,R2,R3].
>
> Here we focus on domain translation between ATAC-seq and RNA-seq measurements. We use human lung adenocarcinoma A549 cells data from [R4]. The dataset contains 1,874 samples of RNA sequences and ATAC sequences. Each data set is split into 1,534 training samples and 340 testing samples following [R1]. As in [R1], we use K-NN accuracy between translated and target domains.  Other settings, metrics follow those in [R1].
>
> Figure 1 in the anonymized link shows the K-NN accuracy of translation from RNA-seq to ATAC-seq of the proposed method and the baseline [R1] (referred to as CM-AE). The results show strong alignment performance of the proposed method, with clear gains due to anchor matching and sparsity regularization. We will add this experiment in the revised version with detailed description of the settings.
>
> [R1] Yang et al. "Multi-domain translation between single-cell imaging and sequencing data using autoencoders." Nature communications, 2021
>
> [R2] Eyring et al, “Unbalancedness in neural monge maps improves unpaired domain translation,” ICLR 2024.
>
> [R3] Amodio et al., “MAGAN: Aligning biological manifolds,” ICML 2018
>
> [R4] Cao et al., “Joint profiling of chromatin accessibility and gene expression in thousands of single cells,” Science 2018.
>
> &nbsp;
>
> **[Why multiple anchors needed?]**
>
> We thank the reviewer for the question. This is an important point worth clarifying. Note that our theorem indicating only 1 pair is needed was derived under the perfect model in Eq. (1), which is validated by the synthetic experiments in Fig. 4. The need for more than one anchor for complex high dimensional data can be attributed to two facts: (i) Theoretical assumptions might not be exactly satisfied by high dimensional real world data, (ii) In practice, nonconvex and min-max optimization Problem (11) cannot be solved exactly/optimally.
>
> &nbsp;
>
> **[Notation and Typos]**
>
> Thanks for your careful reading and pointing out the potentially confusing notation and typo. To avoid confusion, we will change the sentence to
> “Let $\\mathcal{F}\_{i,:} $$= \\{j | (i,j) \\in \\mathcal{F} \\}$ and $\\mathcal{F}{:,j} = \\{j | (i,j) \in \\mathcal{F} \\}$.”
> Here $\\mathcal{F}$ is a set of index pairs, whereas $\\mathcal{X}$ is a set of matrices or vectors. We will further conduct several rounds of proofreading to check for other typos and notation issues for the revised version.

---

> > ### Author Rebuttal · Reviewer_iuhq · 2026-04-02
> >
> > The authors have solved my concerns, so I would like to raise my score.

---

### Official Review · Reviewer_pvLG · 2026-03-16

**Soundness:** 3
**Presentation:** 3
**Significance:** 3
**Originality:** 3
**Overall Recommendation:** 4
**Confidence:** 3

**Summary:**

This paper tackles the problem of non-identifiability in unsupervised DT, which leanrs a mapping between two distributions without paired samples. The authors present an interesting solution to MPA caused content misalignment issue. Main theoretical result shows that under a reasonable structural assumption (Jacobian sparsity), it takes is a single paired anchor sample to force the model to learn the correct mapping. This work is a nice contribution that makes identifiable domain transfer problems easier to solve.

**Compliance With Llm Reviewing Policy:**

Affirmed.

**Key Questions For Authors:**

See above

**Limitations:**

The tasks are too toy.

**Strengths And Weaknesses:**

Strengths And Weaknesses

Strengths:
+ Theory: the unpaired data matching problem has been existing for a long time, it's interesting to show this theoretical result that a single anchor sample can suffice. The proposed solution is also a scalable solution.

+ Writing: The work is technically solid, with clear theoretical claims backed by convincing experiments. The paper is well-written.

Weaknesses:

There are many larger datasets that work on the same problem, could the authors instead try on those and report results? The results on these toy datasets are appreciated, but they do not fully convincingly show the theory capability.

---

> ### Author Rebuttal · Authors · 2026-03-31
>
> **Anonymous link** : [(Click Here)](https://drive.google.com/file/d/1aA_lPh4Qe_ypDaS3raxeceooLjKcIUt0/view?usp=sharing)
>
> &nbsp;
>
> **[More complex Dataset]**
>
> To further show the usefulness of the proposed framework in real-world applications, we run an experiment on a single-cell sequence alignment problem. The task aims to align single cell sequence measurements captured by different modalities (such as RNA-seq and ATAC-seq). The problem stems from the fact that single cell measurement processes are destructive, hence obtaining paired measurements is difficult. This is a core task in biological data analysis and AI4S [R1,R2,R3].
>
> Here we focus on domain translation between ATAC-seq and RNA-seq measurements. We use human lung adenocarcinoma A549 cells data from [R4]. The dataset contains 1,874 samples of RNA sequences and ATAC sequences. Each data set is split into 1,534 training samples and 340 testing samples following [R1]. We use TF-IDF based pre-processing on the count data (fitted on the training set). As in [R1], we use K-NN accuracy between translated and target domains. We use a single linear layer as the translation function trained for 150 epochs with batch size of 64 and learning rate of 0.002. Other settings follow those in [R1].
>
> Figure 1 in the anonymized link shows the K-NN accuracy of translation from RNA-seq to ATAC-seq of the proposed method and the baseline [R1] (referred to as CM-AE). Baseline results are obtained using their github codebase (uhlerlab/cross-modal-autoencoders.git). The results show strong alignment performance of the proposed method compared to the baseline, with clear gains due to anchor matching and sparsity regularization. One can see that a single paired anchor cannot is only effective when combined with the sparsity regularization in our proposed method. We will add this experiment in the revised version with detailed description of the settings.
>
> [R1] Yang et al. "Multi-domain translation between single-cell imaging and sequencing data using autoencoders." Nature communications, 2021
>
> [R2] Eyring et al, “Unbalancedness in neural monge maps improves unpaired domain translation,” ICLR 2024.
>
> [R3] Amodio et al., “MAGAN: Aligning biological manifolds,” ICML 2018
>
> [R4] Cao et al., “Joint profiling of chromatin accessibility and gene expression in thousands of single cells,” Science 2018.

---

### Official Review · Reviewer_E1Jp · 2026-03-17

**Soundness:** 3
**Presentation:** 3
**Significance:** 3
**Originality:** 3
**Overall Recommendation:** 4
**Confidence:** 3

**Summary:**

The authors present a study on identifiability in the context of unsupervised domain transfer (DT). The authors explain how, despite the well-known problem of matching marginals between source and target, there is no unique content-preserving map. This is because measure-preserving automorphisms can distort the matching while preserving marginals. A novel approach is presented for DT Identifiability, where DT is related to sparse non-linear mixing. The authors show that, with a single sample of a paired source-target anchor, the map is identifiable almost everywhere, provided there is a structural sparsity of the Jacobian support of the ground truth transfer map. A learning objective is presented, which includes adversarial matching, anchor loss, invertibility-promoting reconstruction, and scalable masked finite difference surrogate for Jacobian Sparsity. Results have been presented using 2D synthetic data, MNIST to rotated MNIST, and edges to rotated shoes.

**Compliance With Llm Reviewing Policy:**

Affirmed.

**Key Questions For Authors:**

1.	As currently written in Lemma A.3, it is stated for continuous MPAs but the proof uses derivative based and change of variables. A rigorous rework here would improve my assessment without which I do not think the main theorem is fully established yet. If the authors can provide a corrected proof under the assumptions corrected, I am happy to revise my assessment.
2.	Can this theorem to implementation bridge for Eq 14 and 20 be explained in more depth?
3.	In this manuscript, Linv uses a separate reconstruction network, but the code does not use the same estimator as this manuscript. Can this part be explained?
4.	Could you also provide the real image anchor count on Edges to rotated Shoes using 1, 2, 5, 10, along with the content alignment metric, and not just LPIPS?
5.	If the main paper’s claim can be provided regarding the more difficult image benchmark as has been provided for MNIST, it would help me observe this method more clearly.

**Strengths And Weaknesses:**

Soundness: The idea behind the identifiability issue in unsupervised DT by converting the sample wise conditional labels into a single anchor is quite interesting. The proof of Theorem 3.4 heavily relies on Step 3. This is dependent on Lemma A.1, and this is again dependent on Lemma A.3. In this lemma, it is given for a continuous one-dimensional MPA. The proof is based on the derivative. There is a change of variables. It is given that only the non-trivial MPA of the uniform distribution is u -> 1-u. It is given that the proof starts with the continuity of the distribution. Then they use dq/du and claim the derivative to be ±1. I think there might be a typo in Step 3 of the main text. It is given that for non-trivial Π, the set of x satisfying the equation has the measure zero under pz. I think this should be px since the anchor x(l) is chosen based on the source distribution. The strongest experiment is the synthetic task and the MNIST experiment with the rotated MNIST. However, the difficulty of the image benchmark is not as convincing. In fact, there is a singular alignment emphasized in this theorem and manuscript, but in fact, there are 10 anchors being used in the 'Edges to rotated shoes' method. This makes one wonder if this method is only theoretically sound since, in fact, in real-world scenarios, more than 1 anchors are required to be used. Moreover, there is a lack of ablation study in the real image task.

Presentation: The writing is clear and easy to follow at a high level. The figures 1-3 are clear in illustrating the MPA problem and structural sparsity assumption. The authors have a clear presentation of the idea by moving from MPAs to DDM and then to the non-linear unmixing connection. However, the text repeatedly states that the assumptions of the Jacobian structure are mild but then shows in the appendix that these assumptions have significant architectural implications. For example, in the Edges-to-Shoes case, they have changed from a normal instance normalization to a channel normalization due to the fact that instance normalization does not allow the Jacobian of the network to be sparse. This is a significant restriction and should be included in the main text instead of just in the appendix. Also, they have not included CUT (Contrastive Unpaired Translation), which is relevant in the case of unpaired image-to-image translation and content preservation. They have included Park et al. in the related works section but have not included it in the baseline table in the main text.

Significance: The reduction of supervision from the level of the sample-wise conditional labels to the single anchor level is significant, especially as the identifiability in unsupervised domain transfer is a conceptual limit. The convincing experiments on the synthetic setting and rotated MNIST are very good, while the image setting needs stronger evidence.
Originality: The paper is original in the conceptual contribution, as the reuse of the identifiability in domain transfer, now in the form of anchored sparse nonlinear unmixing, is a new angle, and the result on the single anchor identifiability is the most interesting contribution of the paper, in the opinion of the reviewer. The use of the finite difference regularizer is a useful idea for scaling the Jacobian structure to the higher-dimensional setting.

---

> ### Author Rebuttal · Authors · 2026-03-31
>
> **Anonymous link** : [(Click Here)](https://drive.google.com/file/d/1aA_lPh4Qe_ypDaS3raxeceooLjKcIUt0/view?usp=sharing)
>
> **[Proof of Lemma A.3]**
>
> Thank you for the careful reading. You are right that the current proof uses derivatives. In our theory, it is enough to consider differentiable MPAs. To make this precise, we will revise Lemma A.3 to:
>
> “There exists at most one non-identity function $m$ such that $m$ is differentiable and $m_{\\# p_x} = p_x$.”
>
> We will also simplify the proof by removing redundant statements. In particular, we will revise Line 625 to:
>
> “We show that $m_u$ and $m_u'$ are the only differentiable MPAs of $p_u$. Suppose, for contradiction, that there exists another differentiable MPA $q$ of $p_u$, distinct from $m_u$ and $m_u'$. Then there exists $u' \in [0,1]$ such that $\frac{d}{du}q(u)\vert_{u=u'} \neq \pm 1$.”
>
> Regarding $p_z$ and $p_x$, this is a typo: $p_z$ should be $p_x$. In the revision, we will also carefully fix the related typos and any unclear or imprecise statements throughout other proofs.
>
> &nbsp;
>
> **[Deeper Explanation of Sparsity Regularizer (Eqs (14)-(20))]**
>
> In summary, Eq. (20) is a proxy for the exact jacobian sparsity regularization $ E||J_g(x)||_0$. This includes approximation of (i) Jacobian sparsity by Jacobian-vector product sparsity (for computationally efficiency) via Proposition 4.1, (ii) $\ell_0$-norm by $\ell_1$, and (iii) derivatives by finite differences. The justification for these approximations were detailed in Section 4.2 in the manuscript. However, we will add a separate section in the appendix of the revised version for a more thorough explanation.
>
> We are happy to discuss more if the reviewer has more specific questions regarding this part.
>
> &nbsp;
>
> **[L_inv in code]**
>
> The code does use a separate network, denoted by *self.g21* in Line 377 of src/trainer.py.  Note that the proposed method is implemented as a one-sided translation and the reverse direction “g21” is only used for $L_{\rm inv}$. We will refactor the code to make the naming conventions tightly follow the paper’s notations.
>
> &nbsp;
>
> **[Single anchor Theory vs 10 anchors experiment]**
>
> Thank you for this important question. Note that our theorem indicating only 1 pair is needed was derived under the perfect model in Eq. (1) and stated assumptions, which is validated by the synthetic experiment in Fig. 4. The need for more than one anchor for complex high dimensional data can be attributed to two facts: (i) Theoretical assumptions might not be exactly satisfied by high dimensional real world data, (ii) In practice, nonconvex and min-max optimization Problem (11) cannot be solved exactly/optimally.
>
> We will include a more detailed discussion in the revised version.
>
> &nbsp;
>
> **[Number of Anchors]**
>
> Table 1 in the anonymized pdf shows the results on different number of anchors for the edges to rotated shoes. We observe that 1, 2, and 5 aligned samples may be insufficient for this task, possibly due to imperfect optimization and model mismatch as discussed earlier.
>
> Finally, LPIPS is itself a perceptual content-alignment metric, since it measures distance in the feature space of a pretrained vision network. Note that TE is based on raw $\ell_2$ image distance, which is less meaningful for high-dim image translation.
>
> &nbsp;
>
> **[Architectural Restriction]**
>
> Thank you for the suggestion. The assumption does have some structural implications, particularly for normalization, though we did not observe any degradation from channel normalization. Most other architectural choices remain compatible. We will move this discussion to the main text for clarity.
>
> &nbsp;
>
> **[New Image Translation Baseline]**
>
> To address the reviewer’s concern, we added CUT as a baseline. Please check anonymized PDF Table 2 for the results. Note that CUT assumes that patches at the same spatial location contain the same information, which does not hold when the transfer includes geometric transformations, as in our experiments. As a result, CUT performs poorly in this setting (see Fig. 2 in the anonymized pdf).
>
> &nbsp;
>
> **[New Dataset]**
>
> We emphasize that the main contribution of this work is a novel theoretical framework for DT, with experiments primarily intended to validate the theory. Since the claims rely on the stated assumptions, controlled experiments are the most appropriate way to verify the theoretical results.
> That said, we agree that additional experiments help demonstrate broader applicability. To address this, we conducted new experiments on more complex real-world data and, due to time and resource constraints, chose a biomedical single-cell alignment task rather than a high-resolution image experiment. We refer the reviewer to the response to Reviewer pvLG for detailed description of the setting and results.

---

> > ### Author Rebuttal · Reviewer_E1Jp · 2026-04-06
> >
> > My concerns are resolved. I maintain the same score and support the acceptance of this paper.

---

### Decision · Program_Chairs · 2026-04-30

**Decision:**

Accept (regular)

**Comment:**

**Summary and Decision**
While reviewers noted minor concerns regarding the applicability of the method on realistic datasets, the general consensus is that the paper makes a significant and novel contribution to domain transfer research.

The reviewers specifically highlighted the following strengths:
* The writing is clear and easy to follow overall.
* The significance and originality of using a single anchor for identifiability (with Jacobian sparsity assumptions).
* The empirical experiments validate the theoretical claims.

**Rebuttal and Discussion**
During the discussion phase, the authors addressed minor concerns regarding confusions and complex datasets by providing additional explanations and providing another dataset, which satisfied the reviewers. Given the originality of the work, the Area Chair recommends acceptance.

**Final Instructions to Authors**
Please ensure that the promised revisions from the rebuttal—specifically the new dataset and clarified explanations—are incorporated into the camera-ready version.